**METHOD**

# Haplotype-aware diplotyping from noisy long reads

Jana Ebler[1,2,3†], Marina Haukness[4†], Trevor Pesout[4†], Tobias Marschall[1,2*] ⓘ and Benedict Paten[4*]

## Abstract

Current genotyping approaches for single-nucleotide variations rely on short, accurate reads from second-generation sequencing devices. Presently, third-generation sequencing platforms are rapidly becoming more widespread, yet approaches for leveraging their long but error-prone reads for genotyping are lacking. Here, we introduce a novel statistical framework for the joint inference of haplotypes and genotypes from noisy long reads, which we term diplotyping. Our technique takes full advantage of linkage information provided by long reads. We validate hundreds of thousands of candidate variants that have not yet been included in the high-confidence reference set of the Genome-in-a-Bottle effort.

**Keywords:** Computational genomics, Long reads, Genotyping, Phasing, Haplotypes, Diplotypes

## Background

Reference-based genetic variant identification comprises two related processes: genotyping and phasing. Genotyping is the process of determining which genetic variants are present in an individual's genome. A genotype at a given site describes whether both chromosomal copies carry a variant allele, whether only one of them carries it, or whether the variant allele is not present at all. Phasing refers to determining an individual's haplotypes, which consist of variants that lie near each other on the same chromosome and are inherited together. To completely describe all of the genetic variation in an organism, both genotyping and phasing are needed. Together, the two processes are called *diplotyping.*

Many existing variant analysis pipelines are designed for short DNA sequencing reads [1, 2]. Though short reads are very accurate at a per-base level, they can suffer from being difficult to unambiguously align to the

genome, especially in repetitive or duplicated regions [3]. The result is that millions of bases of the reference human genome are not currently reliably genotyped by short reads, primarily in multi-megabase gaps near the centromeres and short arms of chromosomes [4]. While short reads are unable to uniquely map to these regions, long reads can potentially span into or even across them. Long reads have already proven useful for read-based haplotyping, large structural variant detection, and de novo assembly [5–8]. Here, we demonstrate the utility of long reads for more comprehensive genotyping. Due to the historically greater relative cost and higher sequencing error rates of these technologies, little attention has been given thus far to this problem. However, long-read DNA sequencing technologies are rapidly falling in price and increasing in general availability. Such technologies include single-molecule real-time (SMRT) sequencing by Pacific Biosciences (PacBio) and nanopore sequencing by Oxford Nanopore Technologies (ONT), both of which we assess here.

The genotyping problem is related to the task of inferring haplotypes from long-read sequencing data, on which a rich literature and many tools exist [8–14], including our own software WhatsHap [15, 16]. The most common formalization of haplotype reconstruction is the minimum error correction (MEC) problem. The MEC problem seeks to partition the reads by haplotype such that a minimum number of errors need to be corrected in order to make the reads from the same haplotype consistent with each

*Correspondence: t.marschall@mpi-inf.mpg.de; benedict@soe.ucsc.edu
Tobias Marschall and Benedict Paten are joint last authors.
†Jana Ebler, Marina Haukness, and Trevor Pesout contributed equally to this work.
¹Center for Bioinformatics, Saarland University, Saarland Informatics Campus E2.1, 66123 Saarbrücken, Germany
²Max Planck Institute for Informatics, Saarland Informatics Campus E1.4, Saarbrücken, Germany
⁴UC Santa Cruz Genomics Institute, University of California Santa Cruz, 95064 Santa Cruz, CA, USA
Full list of author information is available at the end of the article

other. In principle, this problem formulation could serve to infer genotypes, but in practice, the "all heterozygous" assumption is made: tools for haplotype reconstruction generally assume that a set of heterozygous positions is given as input and exclusively work on these sites.

Despite this general lack of tools, some methods for genotyping using long reads have been proposed. Guo et al. [17] describe a method for long-read single-nucleotide variant (SNV) calling and haplotype reconstruction which identifies an exemplar read at each SNV site that best matches nearby reads overlapping the site. It then partitions reads around the site based on similarity to the exemplar at adjacent SNV sites. However, this method is not guaranteed to discover an optimal partitioning of the reads between haplotypes, and the authors report a comparatively high false discovery rate (15.7%) and false-negative rate (11.0%) for PacBio data of NA12878, which corresponds to an F1 score of only 86.6%. Additionally, two groups are presently developing learning-based variant callers which they show can be tuned to work using long, noisy reads: In a recent preprint, Luo et al. [18] describe a method which uses a convolutional neural network (CNN) to call variants from long-read data, which they report to achieve an F1 score between 94.90 and 98.52%, depending on parametrization (when training on read data from one individual and calling variants on a different individual, see Table 3 of [18]). Poplin et al. [19] present another CNN-based tool, which achieves an F1 score of 92.67% on PacBio data (according to Supplementary Table 3 of [19]). These measures appear promising; however, these methods do not systematically exploit the linkage information between variants provided by long reads. Thus, they do not leverage one of the key advantages of long reads.

For an illustration of the potential benefit of using long reads to diplotype across adjacent sites, consider Fig. 1a. There are three SNV positions shown which are covered by long reads. The gray sequences represent the true haplotype sequences, and reads are colored in blue and red, where the colors correspond to the haplotype which the respective read stems from: the red ones from the upper sequence, and the blue ones from the lower one. Since sequencing errors can occur, the alleles supported by the reads are not always equal to the true ones in the haplotypes shown in gray. Considering the SNVs individually, it would be reasonable to call the first one as A/C, the second one as T/G, and the third one as G/C, since the number of reads supporting each allele is the same. This leads to a wrong prediction for the second SNV. However, if we knew which haplotype each read stems from, that is, if we knew their colors, then we would know that there must be sequencing errors at the second SNV site. Since the reads stemming from the same haplotypes must support the same alleles and there are

discrepancies between the haplotyped reads at this site, any genotype prediction at this locus must be treated as highly uncertain. Therefore, using haplotype information during genotyping makes it possible to detect uncertainties and potentially compute more reliable genotype predictions.

## Contributions

In this paper, we show that for contemporary long read technologies, read-based phase inference can be simultaneously combined with the genotyping process for SNVs to produce accurate diplotypes and to detect variants in regions not mappable by short reads. We show that key to this inference is the detection of linkage relationships between heterozygous sites within the reads. To do this, we describe a novel algorithm to accurately predict diplotypes from noisy long reads that scales to deeply sequenced human genomes.

We then apply this algorithm to diplotype one individual from the 1000 Genomes Project, NA12878, using long reads from both PacBio and ONT. NA12878 has been extensively sequenced and studied, and the Genome in a Bottle Consortium has published sets of high confidence regions and a corresponding set of highly confident variant calls inside these genomic regions [20]. We demonstrate that our method is accurate, that it can be used to confirm variants in regions of uncertainty, and that it allows for the discovery of variants in regions which are unmappable using short DNA read sequencing technologies.

## Results
### A unified statistical framework to infer genotypes and haplotypes

We formulated a novel statistical framework based upon hidden Markov models (HMMs) to analyze long-read sequencing data. In short, we identify potential SNV positions and use our model to efficiently evaluate the bipartitions of the reads, where each bipartition corresponds to assigning each read to one of the individual's two haplotypes. The model ensures that each read stays in the same partition across variants, and hence does not "switch haplotypes," something which is key to exploiting the inherent long range information. Based on the read support of each haplotype at each site, the model determines the likelihood of the bipartition. By using the forward-backward algorithm, we pursue "global" diplotype inference over whole chromosomes, a process that yields genotype predictions by determining the most likely genotype at each position, as well as haplotype reconstructions. In contrast to panel-based methods, like the Li-Stephens model [21], our method relies on read data instead of using knowledge of existing haplotypes. In Fig. 1b, we give a conceptual overview of our approach and describe it in more detail in the "Methods" section.

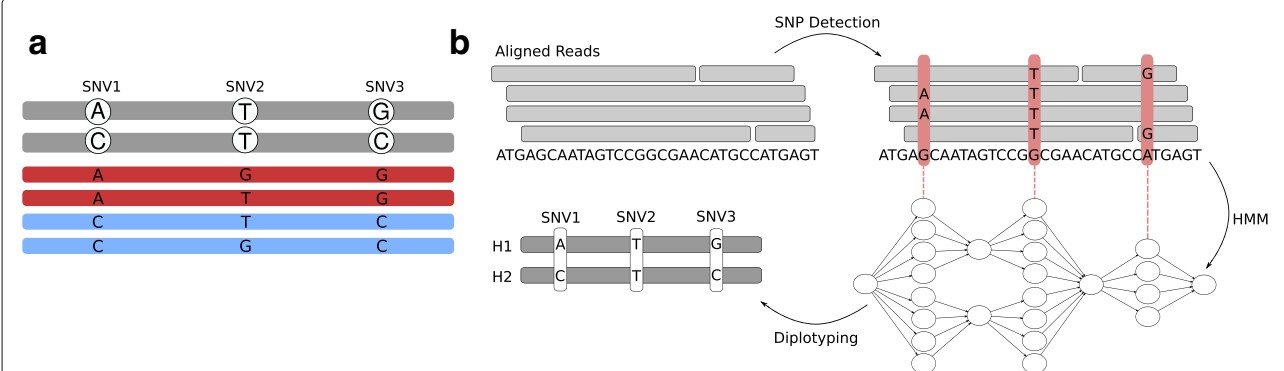

**Fig. 1** Motivation and overview of diplotyping. **a** Gray sequences illustrate the haplotypes; the reads are shown in red and blue. The red reads originate from the upper haplotype, the blue ones from the lower. Genotyping each SNV individually would lead to the conclusion that all of them are heterozygous. Using the haplotype context reveals uncertainty about the genotype of the second SNV. **b** Clockwise starting top left: first, sequencing reads aligned to a reference genome are given as input; second, the read alignments are used to nominate candidate variants (red vertical bars), which are characterized by the differences to the reference genome; third, a hidden Markov model (HMM) is constructed where each candidate variant gives rise to one "row" of states, representing possible ways of assigning each read to one of the two haplotypes as well as possible genotypes (see the "Methods" section for details); forth, the HMM is used to perform *diplotyping*, i.e., we infer genotypes of each candidate variant as well as how the alleles are assigned to haplotypes

Adding robustness to our analysis, we provide two independent software implementations of our model: one is made available as an extension to WhatsHap [16, 22] and the other is a from-scratch implementation called Margin-Phase. While the core algorithmic ideas are the same, MarginPhase and WhatsHap differ primarily in their specialized preprocessing steps, with the former being developed to work primarily with nanopore data and the latter developed to work primarily with PacBio (although both can work with either). The MarginPhase workflow includes an alignment summation step described in the "Allele supports" section whereas WhatsHap performs a local realignment around analyzed sites explained in the "Allele detection" section.

### Data preparation and evaluation

To test our methods, we used sequencing data for NA12878 from two different long-read sequencing technologies. NA12878 is a participant from the 1000 Genomes Project [2] who has been extensively sequenced and analyzed. This is the only individual for whom there is both PacBio and Nanopore sequencing reads publicly available. We used Oxford Nanopore reads from Jain et al. [7] and PacBio reads from the Genome in a Bottle Consortium [23]. Both sets of reads were aligned to GRCh38 with `minimap2`, a mapper designed to align error-prone long reads [24] (version 2.10, using default parameters for PacBio and Nanopore reads, respectively).

To ensure that any variants we found were not artifacts of misalignment, we filtered out the reads flagged as secondary or supplementary, as well as reads with a mapping quality score less than 30. Genome-wide, this left approximately 12 million Nanopore reads and 35 million PacBio reads. The Nanopore reads had a median depth of $37\times$ and median length of 5950 bp, including a set of ultra-long reads with lengths up to 900 kb. The PacBio reads had a median depth of $46\times$ and median length of 2600 bp.

To validate the performance of our methods, we used callsets from Genome in a Bottle's (GIAB) benchmark small variant calls v3.3.2 [20]. First, we compared against GIAB's set of high confidence calls, generated by a consensus algorithm spanning multiple sequencing technologies and variant calling programs. The high confidence regions associated with this callset exclude structural variants, centromeres, and heterochromatin. We used this to show our method's accuracy in well-understood and easy-to-map regions of the genome.

We also analyzed our results compared to two more expansive callsets, which cover a larger fraction of the genome, that were used in the construction of GIAB's high confidence variants, one made by GATK Haplotype-Caller v3.5 (GATK/HC, [1]) and the other by Freebayes 0.9.20 [25], both generated from a $300\times$ PCR-free Illumina sequencing run [20].

### Evaluation statistics

We compute the precision and recall of our callsets using the tool `vcfeval` from Real Time Genomics [26] (version 3.9) in order to analyze our algorithm's accuracy of variant detection between our query callsets and a baseline truth set of variants. All variants that identically match between the truth and query callsets (meaning they share the same genomic position, alleles, and genotype) are

considered true positive calls. Calls that do not match any variants in the truth callset are false negatives and truth callset variants that are not matched in our callset are false positives.

In order to evaluate the ability of our algorithm to genotype a provided set of variant positions, we compute the genotype concordance. Here, we take all correctly identified variant sites (correct genomic position and alleles), compare the genotype predictions (homozygous or heterozygous) made by our method to the corresponding truth set genotypes, and compute the fraction of correct genotype predictions. This enables us to analyze how well the genotyping component of our model performs regardless of errors arising from wrongly called variant sites in the detection stage of the algorithm.

We evaluate the phasing results by computing the switch error rate between the haplotypes our algorithms predict and the truth set haplotypes. We take all variants into account that were correctly genotyped as heterozygous in both our callset and the truth set. Switch errors are calculated by counting the number of times a jump from one haplotype to the other is necessary within a phased block in order to reconstruct the true haplotype sequence [16].

We restrict all analysis to SNVs, not including any short insertions or deletions. This is due to the error profile of both PacBio and Nanopore long reads, for which erroneous insertions and deletions are the most common type of sequencing error by far, particularly in homopolymers [27, 28].

### Comparison to short read variant callers

We explored the suitability of the current state-of-the-art callers for short reads to process long-read data (using default settings) but were unsuccessful. The absence of base qualities in the PacBio data prevented any calling; for Nanopore data, FreeBayes was prohibitively slow and neither Platypus nor GATK/HC produced any calls.

### Long read coverage

We determined the regions where long and short reads can be reliably mapped to the human genome for the purpose of variant calling, aiming to understand if long reads could potentially make new regions accessible. In Fig. 2, various coverage metrics for short and long reads are plotted against different genomic features, including those known for being repetitive or duplicated. These metrics are described below.

The callsets on the Illumina data made by GATK/HC and FreeBayes come with two BED files describing where calls were made with some confidence. The first, described in Fig. 2 as *Short Read Mappable*, was generated using GATK CallableLoci v3.5 and includes regions where there is (a) at least a read depth of 20 and (b) at

most a depth of twice the median depth, only including reads with mapping quality of at least 20. This definition of callable only considers read mappings.

The second, described as *GATK Callable*, was generated from the GVCF output from GATK/HC by excluding the areas with genotype quality less than 60. This is a more sophisticated definition of callable as it reflects the effects of homopolymers and tandem repeats. We use these two BED files in our analysis of how short and long reads map differently in various areas of the genome.

For long reads, we show four coverage statistics. The entries marked as "mappable" describe the areas where there is at least one high-quality long-read mapping (*PacBio Mappable*, *Nanopore Mappable*, and *Long Read Mappable* for regions where at least one of the sequencing technologies mapped). The *Long Read Callable* entries cover the regions where our methods should be able to call variants due to having a sufficient depth of read coverage. In these regions, both sequencing technologies had a minimum read depth of 20 and a maximum of twice the median depth (this is similar to the GATK CallableLoci metric, although made from BAMs with significantly less read depth).

Figure 2 shows that in almost all cases, long reads map to a higher fraction of the genome than short reads map to. For example, nearly half a percent of the whole genome is mappable by long reads but not short reads. Long reads also map to 1% more of the exome, and 13% more of segmental duplications. Centromeres and tandem repeats are outliers to this generalization, where neither PacBio nor Nanopore long reads cover appreciably more than Illumina short reads.

### Comparison against high confidence truth set

To validate our method, we first analyzed the SNV detection and genotyping performance of our algorithm using the GIAB high confidence callset as a benchmark. All variants reported in these statistics fall within both the GIAB high confidence regions and regions with a read depth between 20 and twice the median depth.

### Variant detection

Figure 3 (top) shows precision and recall of WhatsHap run on PacBio data and MarginPhase on Oxford Nanopore data, which gives the best performance for these two data types (see Additional file 1: Figure S1 for the results for WhatsHap on ONT and MarginPhase on PacBio). On PacBio reads, WhatsHap achieves a precision of 97.9% and recall of 96.3%. On Nanopore reads, MarginPhase achieves a precision of 76.9% and a recall of 80.9%. We further stratify the performance of our methods based on the variant type. For homozygous variants, WhatsHap on PacBio data has a precision of 98.3% and a recall of 99.3%, MarginPhase on Nanopore data has a precision of

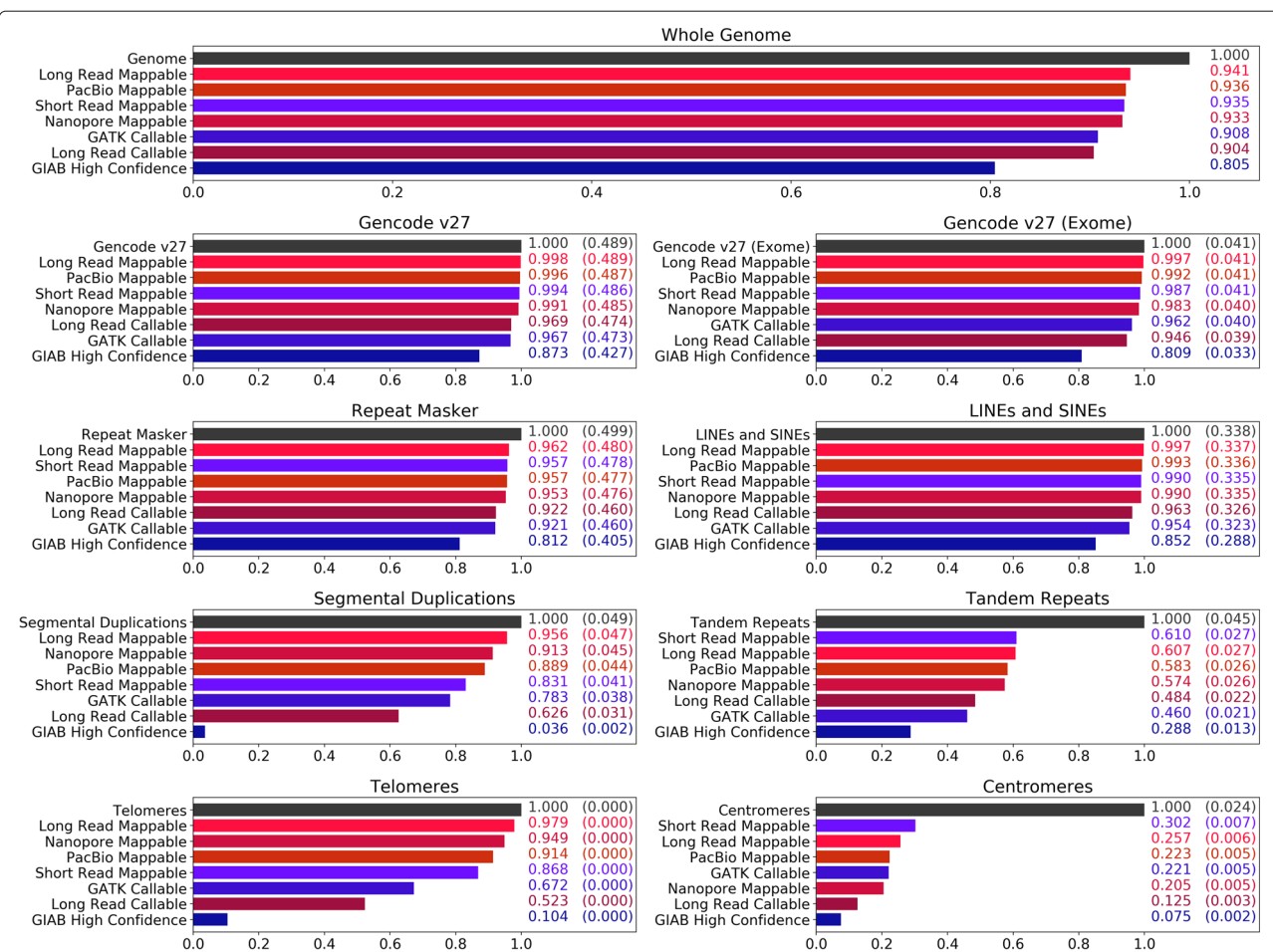

**Fig. 2** Reach of short read and long read technologies. The callable and mappable regions for NA12878 spanning various repetitive or duplicated sequences on GRCh38 are shown. Feature locations are determined based on BED tracks downloaded from the UCSC Genome Browser [48]. Other than the Gencode regions [49, 50], all features are subsets of the Repeat Masker [51] track. Four coverage statistics for long reads (shades of red) and three for short reads (shades of blue) are shown. The labels "PacBio Mappable" and "Nanopore Mappable" describe areas where at least one primary read with GQ ≥ 30 has mapped, and "Long Read Mappable" describes where this is true for at least one of the long read technologies. "Long Read Callable" describes areas where both read technologies have coverage of at least 20 and less than twice the median coverage. "GIAB High Confidence," "GATK Callable," and "Short Read Mappable" are the regions associated with the evaluation callsets. For the feature-specific plots, the numbers on the right detail coverage over the feature and coverage over the whole genome (parenthesized)

99.3% and a recall of 84.5%. For heterozygous variants, WhatsHap on PacBio data has a precision of 96.8% and a recall of 93.8%; MarginPhase on Nanopore data has a precision of 66.5% and a recall of 78.6%. The high error rate of long reads contributes to the discrepancy in the performance between homozygous and heterozygous variant detection, making it more difficult to distinguish the read errors from the alternate allele for heterozygous variants. In Section 5 of Additional file 1, we further discuss the precision and recall as a function of read depth, and we report more performance based on variant type in Section 6.

Long reads have the ability to access regions of the genome inaccessible to short reads ("Long read coverage" section). To explore the potential of our approach

to contribute to extending gold standard sets, such as the one produced by the GIAB effort, we produced a combined set of variants which occur in both the calls made by WhatsHap on the PacBio reads and Margin-Phase on the Nanopore data, where both tools report the same genotype. This improves the precision inside the GIAB high confidence regions to 99.7% with a recall of 78.7%. In further analysis, we refer to this combined variant set as *Long Read Variants*. It reflects a high precision subset of variants validated independently by both sequencing technologies. While data from both technologies are usually not available for the same sample in routine settings, such a call set can be valuable for curating variants on well-studied individuals such as NA12878.

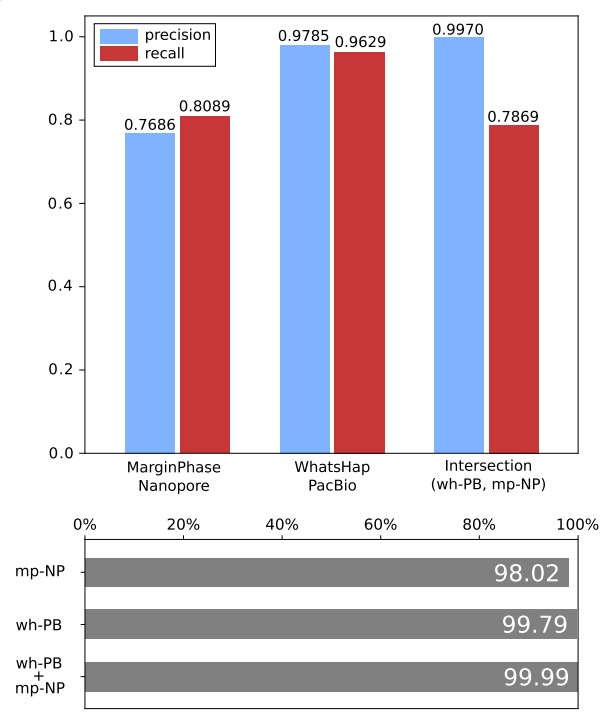

**Fig. 3** Precision and recall of MarginPhase on Nanopore and WhatsHap on PacBio datasets in GIAB high confidence regions. Genotype concordance (bottom) (wrt. GIAB high confidence calls) of MarginPhase (mp, top) on Nanopore and WhatsHap (wh, middle) on PacBio (PB). Furthermore, genotype concordance for the intersection of the calls made by WhatsHap on the PacBio and MarginPhase on the Nanopore reads is shown (bottom)

### Genotyping

In order to further analyze the quality of the genotype predictions of our methods (heterozygous or homozygous), we computed the genotype concordance (defined in the "Data preparation and evaluation" section) of our callsets with respect to the GIAB ground truth inside of the high confidence regions. Figure 3 (bottom) shows the results. On the PacBio data, WhatsHap obtains a genotype concordance of 99.79. On the Nanopore data, MarginPhase obtains a genotype concordance of 98.02. Considering the intersection of the WhatsHap calls on PacBio, and MarginPhase calls on Nanopore data (i.e., the *Long Read Variants* set), we obtain a genotype concordance of 99.99%. We detail the genotype performances for different thresholds on the genotype quality scores that our methods report for each variant call (Additional file 1: Section 7).

### Phasing

In addition to genotyping variants, MarginPhase and WhatsHap can also phase them. We evaluated the results of both methods by computing switch error rates (defined in thr "Data preparation and evaluation" section) inside the GIAB high-confidence regions for correctly located and genotyped GIAB truth set variants. We computed the

switch error rate of MarginPhase on Nanopore and WhatsHap on PacBio reads. For both datasets, we achieved a low switch error rate of 0.17%. In Additional file 1: Table S1, corresponding per-chromosome switch error rates are given.

### Cutting and downsampling reads

Our genotyping model incorporates haplotype information into the genotyping process by using the property that long sequencing reads can cover multiple variant positions. Therefore, one would expect the genotyping results to improve as the length of the provided sequencing reads increases.

In order to examine how the genotyping performance depends on the length of the sequencing reads and the coverage of the data, the following experiment was performed using the WhatsHap implementation. The data was downsampled to average coverages $10\times, 20\times, 25\times$, and $30\times$. All SNVs inside of the high confidence regions in the GIAB truth set were re-genotyped from each of the resulting downsampled read sets, as well as from the full coverage data sets. Two versions of the genotyping algorithm were considered. First, the full-length reads as given in the BAM files were provided to WhatsHap. Second, in an additional step prior to genotyping, the aligned sequencing reads were cut into shorter pieces such that each resulting fragment covered at most two variants. Additionally, we cut the reads into fragments covering only one variant position. The genotyping performances of these genotyping procedures were finally compared by determining the amount of incorrectly genotyped variants.

Figure 4 shows the results of this experiment for the PacBio data. The genotyping error increases as the length of reads decreases. Especially at lower coverages, the genotyping algorithm benefits from using the full length reads, which leads to much lower genotyping errors compared to using the shorter reads that lack information of neighboring variants. For the Nanopore reads, the results were very similar (Additional file 1: Figure S2). In general, the experiment demonstrates that incorporating haplotype information gained from long reads does indeed improve the genotyping performance. This is especially the case at low coverages, since here, the impact of sequencing errors on the genotyping process is much higher. Computing genotypes based on bipartitions of reads that represent possible haplotypes of the individual helps to reduce the number of genotyping errors, because it makes it easier to detect sequencing errors in the given reads.

### Callset consensus analysis

Call sets based on long reads might contribute to improving benchmark sets such as the GIAB truth set. We

analyze a call set created by taking the intersection of the variants called by WhatsHap on PacBio reads and MarginPhase on Nanopore reads, which leaves variants that were called identically between the two sets. In Fig. 5, we further dissect the relation of this intersection callset, which we call *Long Read Variants*, to the GIAB truth set, as well as its relation to the callsets from GATK HaplotypeCaller and FreeBayes, which both contributed to the GIAB truth set.

Figure 5a reveals that 404,994 variants in our *Long Read Variants* callset were called by both the GATK Haplotype Caller and FreeBayes, yet are not in the GIAB truth set. To gather additional support for the quality of these calls, we consider two established quality metrics: the transition/transversion ratio (Ti/Tv) and the heterozygous/non-ref homozygous ratio (Het/Hom) [29]. The Ti/Tv ratio of these variants is 2.09, and the Het/Hom ratio is 1.31. These ratios are comparable to those of the GIAB truth set, which are 2.10 and 1.55, respectively. An examination of the Platinum Genomes benchmark set [30], an alternative to GIAB, reveals 78,493 such long-read validated variants outside of their existing truth set.

We hypothesized that a callset based on long reads is particularly valuable in the regions that were previously difficult to characterize. To investigate this, we separately examined the intersections of our *Long Read Variants* callset with the two short-read callsets both inside the GIAB high confidence regions and outside of them, see Fig. 5b and c, respectively. These Venn diagrams clearly indicate that the concordance of GATK and FreeBayes was indeed substantially higher in high confidence regions than outside. An elevated false-positive rate of the short-read callers outside the high confidence regions is a plausible explanation for this observation. Interestingly,

the fraction of calls concordant between FreeBayes and GATK for which we gather additional support is considerably lower outside the high confidence regions. This is again compatible with an increased number of false positives in the short-read callsets, but we emphasize that these statistics should be interpreted with care in the absence of a reliable truth set for these regions.

## Candidate novel variants

To demonstrate that our method allows for variant calling on more regions of the genome than short-read variant calling pipelines, we have identified 15,498 variants which lie outside of the *Short Read Mappable* area, but inside the *Long Read Callable* regions. These variants therefore fall within the regions in which there is a sequencing depth of at least 10 and not more than 2 times the median depth for both long-read sequencing technologies, yet the regions are unmappable by short reads. We determined that 4.43 Mb of the genome are only mappable by long reads in this way.

Table 1 provides the counts of all variants found in each of the regions from Fig. 2, as well as the counts for candidate novel variants, among the different types of genomic features described in "Long read coverage" section. Over two thirds of the candidate variants occurred in the repetitive or duplicated regions described in the UCSC Genome Browser's repeatMasker track. The transition/transversion ratio (Ti/Tv) of NA12878's 15,498 candidate variants is 1.64, and the heterozygous/homozygous ratio (Het/Hom) of these variants is 0.31. Given that we observe 1 candidate variant in every 325 haplotype bases of the 4.43 Mb of the genome only mappable by long reads, compared to 1 variant in every 1151 haplotype bases in the GIAB truth set on the whole genome, these candidate variants exhibit a 3.6× increase in the haplotype variation rate.

## Runtimes

Whole-genome variant detection using WhatsHap took 166 CPU hours on PacBio reads, of which genotyping took 44 h. Per chromosome, a maximum of 4.2 GB of memory was required for genotyping, and additionally, at most 2.6 GB was needed for phasing. The MarginPhase implementation took 1550 CPU hours on ONT data, broken down into 330 h for diplotyping and 1220 h for read realignment (described in the "Allele supports" section). The MarginPhase workflow breaks the genome into 2-Mb overlapping windows, and on each of these windows, MarginPhase required on average 22.6 GB of memory, and a maximum of 30.2 GB.

We found that the time-consuming realignment step significantly improved the quality of the ONT results and attribute this as the major cause of the difference in runtimes. Furthermore, the methods employed to the

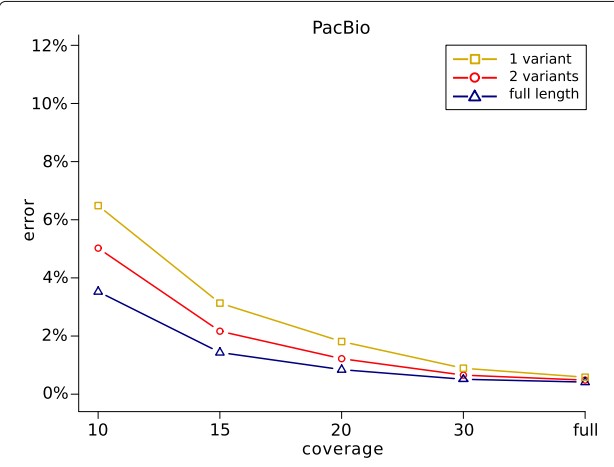

**Fig. 4** Genotyping errors (with respect to GIAB calls) as a function of coverage. The full length reads were used for genotyping (blue), and additionally, reads were cut such as to cover at most two variants (red) and one variant (yellow)

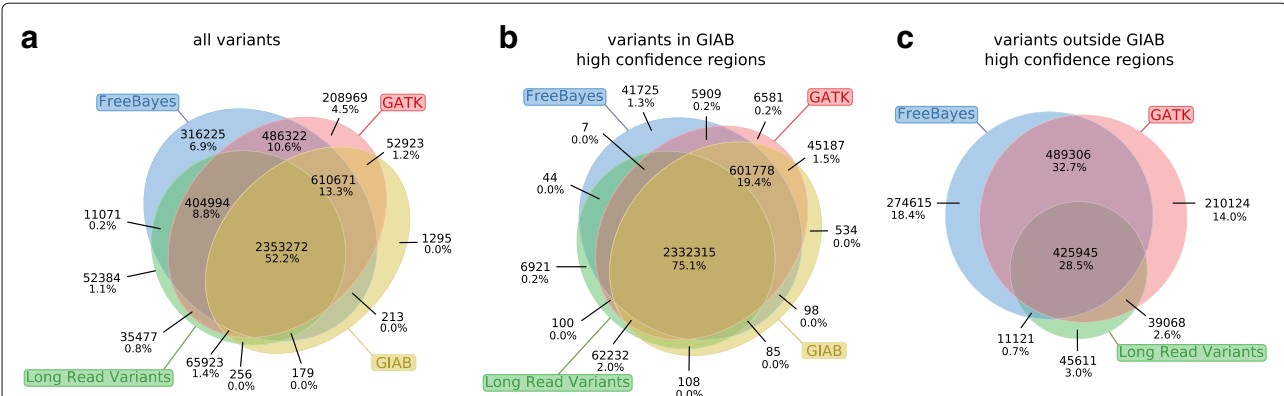

**Fig. 5** Confirming short-read variants. We examine all distinct variants found by our method, GIAB high confidence, GATK/HC, and FreeBayes. Raw variant counts appear on top of each section, and the percentage of total variants is shown at the bottom. **a** All variants. **b** Variants in GIAB high-confidence regions. **c** Variants outside GIAB high-confidence regions

find candidate sites differ between the implementations. WhatsHap performs genotyping and phasing in two steps, whereas MarginPhase handles them simultaneously after filtering out the sites that are likely homozygous (in the case of ONT data, this is between 98 and 99% of sites). The filtration heuristic used during our evaluation resulted in MarginPhase analyzing roughly 10× the number of sites than WhatsHap, increasing the runtime and memory usage.

## Discussion

We introduce a novel statistical model to unify the inference of genotypes and haplotypes from long (but noisy) third-generation sequencing reads, paving the way for genotyping at intermediate coverage levels. We emphasize that our method operates at coverage levels that preclude the possibility of performing a de novo genome assembly, which, until now, was the most common use of long-read

**Table 1** Distribution of candidate novel variants across different regions of interest

|  | All variants | Novel variant candidates |
| --- | --- | --- |
| Total | 2,923,556 | 14,118 |
| Gencode v27 (ALL) | 1,483,947 | 5,093 |
| Gencode v27 exome | 92,268 | 321 |
| Repeat Masker | 1,592,193 | 9,954 |
| LINEs | 687,989 | 4,808 |
| SINEs | 335,181 | 953 |
| Segmental duplications | 161,588 | 4,838 |
| Tandem repeats | 104,753 | 5,944 |
| Centromeres | 1,037 | 28 |
| Telomeres | 0 | 0 |

All variants refers to the variants in the *Long Read Variants* set, and Novel Variant Candidates are those described in "Candidate novel variants" section

data. Furthermore, we note that unlike the approaches using a haplotype reference panel of a population for statistical phasing and/or imputation [31], our approach only uses sequencing data from the individual; hence, its performance does not rely on the allele frequency within a population.

Our method is based on a hidden Markov model that partitions long reads into haplotypes, which we found to improve the quality of variant calling. This is evidenced by our experiment in cutting and downsampling reads, where reducing the number of variants spanned by any given read leads to decreased performance at all levels of read coverage. Therefore, our method is able to translate the increased read lengths of third generation platforms into increased genotyping performance for these noisy long reads.

Our analysis of the methods against a high confidence truth set in high confidence regions shows false discovery rates (corresponding to one minus precision) between 3 and 6% for PacBio and between 24 and 29% for Nanopore. However, when considering a conservative set of variants confirmed by both long read technologies, the false discovery rate drops to around 0.3%, comparable with contemporary short-read methods in these regions.

In analyzing the area of the genome with high-quality long-read mappings, we found roughly a half a percent of the genome (approximately 15 Mb) that is mappable by long reads but not by short reads. This includes 1% of the human exome, as well as over 10% of segmental duplications. Even though some of these areas have low read counts in our experimental data, the fact that they have high-quality mappings means that they should be accessible with sufficient sequencing. We note that this is not the case for centromeric regions, where Illumina reads were able to map over twice as much as we found in our PacBio data. This may be a result of the low quality in long reads

preventing them from uniquely mapping to these areas with an appreciable level of certainty.

We demonstrate that our callset has expected biological features, by showing that over our entire set of called variants, the Ti/Tv and Het/Hom ratios were similar to those reported by the truth set. The Ti/Tv ratio of 2.18 is slightly above the 2.10 reported in the GIAB callset, and the Het/Hom ratio of 1.36 is slightly lower than the 1.55 found in the GIAB variants. In the 15,498 novel variant candidates produced by our method in regions unmappable by short reads, the Ti/Tv ratio of 1.64 is slightly lower than that of the truth set. This is not unexpected as gene-poor regions such as these tend to have more transversions away from C:G pairs [32]. We also observe that the Het/Hom ratio dropped to 0.31, which could be due to the systematic biases in our callset or in the reference genome. The rate of variation in these regions was also notably different than in the high confidence regions, where we find three variants per thousand haplotype bases ($3.6\times$ the rate in high confidence regions). A previous study analyzing NA12878 [33] also found an elevated variation rate in the regions where it is challenging to call variants, such as low-complexity regions and segmental duplications. The study furthermore found clusters of variants in these regions, which we also observe.

The high precision of our set containing the intersection of variants called on Nanopore reads and variants called on PacBio reads makes it useful as strong evidence for confirming existing variant calls. As shown in the read coverage analysis, in both the GIAB and Platinum Genomes efforts many regions could not be called with high confidence. In the regions excluded from GIAB, we found around 400,000 variants using both Nanopore and PacBio reads with our methods, which were additionally confirmed by 2 other variant callers, FreeBayes and GATK/HC, on Illumina reads. Given the extensive support of these variants from multiple sequencing technologies and variant callers, these 400,000 variants are good candidates for addition to the GIAB truth set. Expansion of benchmark sets to harder-to-genotype regions of the human genome is generally important for the development of more comprehensive genotyping methods, and we plan to work with these efforts to use our results.

## Conclusions

Variant calling with long reads is difficult because they are lossy and error-prone, but the diploid nature of the human genome provides a means to partition reads to lessen this effect. We exploit the fact that reads spanning heterozygous sites must share the same haplotype to differentiate read errors from true variants. We provide two implementations of this method in two long-read variant callers, and while both implementations can be run on either sequencing technology, we currently recommend

that MarginPhase is used on ONT data and that WhatsHap is used on PacBio data.

One way we anticipate improvement to our method is by incorporating methylation data. Hidden Markov models have been used to produce methylation data for ONT reads using the underlying signal information [34, 35]. As shown by the read-cutting experiment, the amount of heterozygous variants spanned by each read improves our method's accuracy. We predict that the inclusion of methylation into the nucleotide alphabet will increase the amount of observable heterozygosity and therefore further improve our ability to call variants. Work has begun to include methylation probabilities into our method.

The long-read genotyping work done by Luo et al. [18] using CNNs does not account for haplotype information. Partitioning reads into haplotypes as a preprocessing step (such as our method does) may improve the CNN's performance; we think this is an interesting avenue of exploration.

Further, our method is likely to prove useful for future combined diplotyping algorithms when both genotype and phasing is required, for example, as may be used when constructing phased diploid de novo assemblies [36, 37] or in future hybrid long/short-read diplotyping approaches. Therefore, we envision the statistical model introduced here to become a standard tool for addressing a broad range of challenges that come with long-read sequencing of diploid organisms.

## Methods

We describe a probabilistic model for diplotype inference, and in this paper use it, primarily, to find maximum posterior probability genotypes. The approach builds upon the WhatsHap approach [22] but incorporates a full probabilistic allele inference model into the problem. It has similarities to that proposed by Kuleshov [38], but we here frame the problem using Hidden Markov models (HMMs).

### Alignment matrix

Let $\mathbf{M}$ be an alignment matrix whose rows represent sequencing *reads* and whose columns represent genetic *sites*. Let $m$ be the number of rows, let $n$ be the number of columns, and let $\mathbf{M}_{i,j}$ be the $j$th element in the $i$th row. In each column, let $\Sigma_j \subset \Sigma$ represent the set of possible *alleles* such that $\mathbf{M}_{i,j} \in \Sigma_j \cup \{-\}$, the "$-$" gap symbol representing a site at which the read provides no information. We assume no row or column is composed only of gap symbols, an uninteresting edge case. An example alignment matrix is shown in Fig. 6. Throughout the following, we will be informal and refer to a row $i$ or column $j$, being clear from the context whether we are referring to the row or column itself or the coordinate.

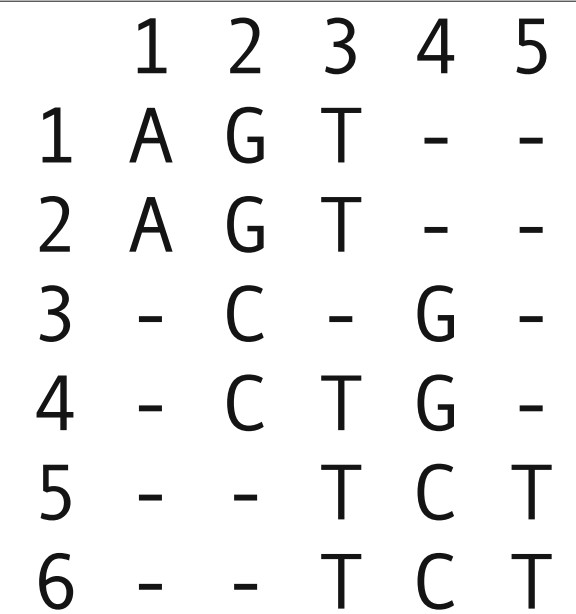

**Fig. 6** Alignment matrix. Here, the alphabet of possible alleles is the set of DNA nucleotides, i.e., $\Sigma = \{A, C, G, T\}$

## Genotype inference problem overview

A diplotype $H = (H^1, H^2)$ is a pair of haplotype (segments); a *haplotype (segment)* $H^k = H_1^k, H_2^k, \ldots, H_n^k$ is a sequence of length $n$ whose elements represents alleles such that $H_j^k \in \Sigma_j$. Let $B = (B^1, B^2)$ be a bipartition of the rows of $\mathbf{M}$ into two parts (sets): $B^1$, the first part, and $B^2$, the second part. We use bipartitions to represent which haplotypes the reads came from, of the two in a genome. By convention, we assume that the first part of $B$ are the reads arising from $H^1$ and the second part of $B$ are the reads arising from $H^2$.

The problem we analyze is based upon a probabilistic model that essentially represents the (weighted) minimum error correction (MEC) problem [39, 40], while modeling the evolutionary relationship between the two haplotypes and so imposing a cost on bipartitions that create differences between the inferred haplotypes.

For a bipartition $B$, and making an i.i.d. assumption between sites in the reads:

$$P(H|B, \mathbf{M}) = \prod_{j=1}^{n} \sum_{Z_j \in \Sigma_j} P\left(H_j^1 | B^1, Z_j\right) P\left(H_j^2 | B^2, Z_j\right) P(Z_j)$$

Here, $P(Z_j)$ is the prior probability of the ancestral allele $Z_j$ of the two haplotypes at column $j$, by default we can use a simple flat distribution over ancestral alleles (but see below). The posterior probability $P(H_j^k | B^k, Z_j) =$

$$\frac{P\left(H_j^k | Z_j\right) \prod_{\{i \in B^k : \mathbf{M}_{i,j} \neq -\}} P\left(\mathbf{M}_{i,j} | H_j^k\right)}{\sum_{Y_j \in \Sigma_j} P(Y_j | Z_j) \prod_{\{i \in B^k : \mathbf{M}_{i,j} \neq -\}} P(\mathbf{M}_{i,j} | Y_j)}$$

for $k \in \{1, 2\}$, where the probability $P\left(H_j^k | Z_j\right)$ is the probability of the haplotype allele $H_j^k$ given the ancestral allele $Z_j$. For this, we can use a continuous time Markov model for allele substitutions, such as Jukes and Cantor [41], or some more sophisticated model that factors the similarities between alleles (see below). Similarly, $P\left(\mathbf{M}_{i,j} | H_j^k\right)$ is the probability of observing allele $\mathbf{M}_{i,j}$ in a read given the haplotype allele $H_j^k$.

The genotype inference problem we consider is finding for each site:

$$\underset{\left(H_j^1, H_j^2\right)}{\arg \max} P\left(H_j^1, H_j^2 | \mathbf{M}\right) = \underset{\left(H_j^1, H_j^2\right)}{\arg \max} \sum_{B} P\left(H_j^1, H_j^2 | B, \mathbf{M}\right)$$

i.e., finding the genotype $\left(H_j^1, H_j^2\right)$ with maximum posterior probability for a generative model of the reads embedded in $\mathbf{M}$.

### A graphical representation of read partitions

For column $j$ in $\mathbf{M}$, row $i$ is *active* if the first non-gap symbol in row $i$ occurs at or before column $j$ and the last non-gap symbol in row $i$ occurs at or after column $j$. Let $A_j$ be the set of active rows of column $j$. For column $j$, row $i$ is *terminal* if its last non-gap symbol occurs at column $j$ or $j = n$. Let $A_j'$ be the set of active, non-terminal rows of column $j$.

Let $B_j = \left(B_j^1, B_j^2\right)$ be a bipartition of $A_j$ into the first part $B_j^1$ and a second part $B_j^2$. Let $\mathbf{B_j}$ be the set of all possible such bipartitions of the active rows of $j$. Similarly, let $C_j = \left(C_j^1, C_j^2\right)$ be a bipartition of $A_j'$ and $\mathbf{C_j}$ be the set of all possible such bipartitions of the active, non-terminal rows of $j$.

For two bipartitions $B = (B^1, B^2)$ and $C = (C^1, C^2)$, $B$ is *compatible* with $C$ if the subset of $B^1$ in $C^1 \cup C^2$ is a subset of $C^1$, and, similarly, the subset of $B^2$ in $C^1 \cup C^2$ is a subset of $C^2$. Note this definition is symmetric and reflexive, although not transitive.

Let $G = (V_G, E_G)$ be a directed graph. The vertices $V_G$ are the set of bipartitions of both the active rows and the active, non-terminal rows for all columns of $\mathbf{M}$ and a special *start* and *end* vertex, i.e., $V_G = \{\text{start, end}\} \cup (\bigcup_j \mathbf{B_j} \cup \mathbf{C_j})$. The edges $E_G$ are a subset of compatibility relationships, such that (1) for all $j$, there is an edge $(B_j \in \mathbf{B_j}, C_j \in \mathbf{C_j})$ if $B_j$ is compatible with $C_j$; (2) for all $1 \leq j < n$, there is an edge $(C_j \in \mathbf{C_j}, B_{j+1} \in \mathbf{B_{j+1}})$ if $C_j$ is compatible with $B_{j+1}$; (3) there is an edge from the start vertex to each member of $\mathbf{B_1}$; and (4) there is an edge from each member of $\mathbf{B_n}$ to the end vertex (note that $\mathbf{C_n}$ is empty and so contributes no vertices to $G$). Figure 7 shows an example graph.

The graph $G$ has a large degree of symmetry and the following properties are easily verified:

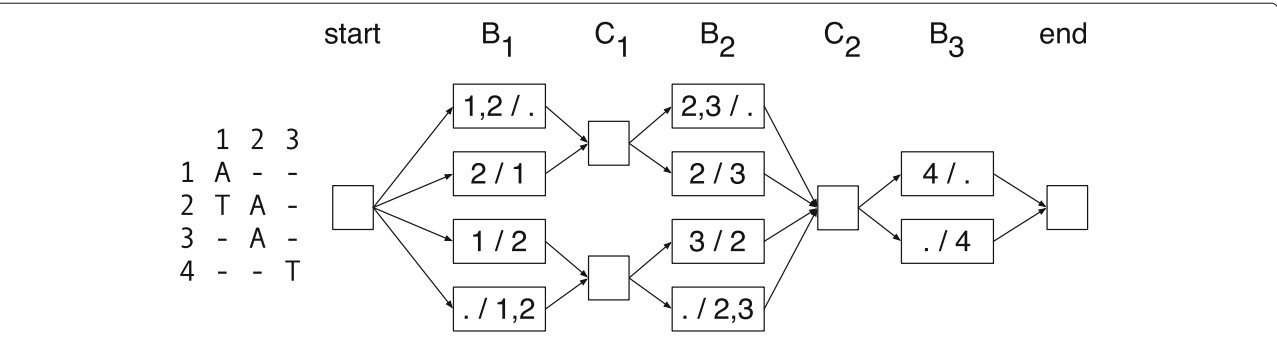

**Fig. 7** Example graph. Left—an alignment matrix. Right—the corresponding directed graph representing the bipartitions of active rows and active non-terminal rows, where the labels of the nodes indicate the partitions, e.g., "1,2 / ." is shorthand for $A = (\{1, 2\}, \{\})$

- For all $j$ and all $B_j \in \mathbf{B_j}$, the indegree and outdegree of $B_j$ is 1.
- For all $j$, the indegree of all members of $\mathbf{C_j}$ is equal.
- Similarly, for all $j$, the outdegree of all members of $\mathbf{C_j}$ is equal.

Let the *maximum coverage*, denoted *maxCov*, be the maximum cardinality of a set $A_j$ over all $j$. By definition, maxCov $\leq$ $m$. Using the above properties it is easily verified that (1) the cardinality of $G$ (number of vertices) is bounded by this maximum coverage, being less than or equal to $2 + (2n - 1)2^{\mathrm{maxCov}}$ and (2) the size of $G$ (number of edges) is at most $2n2^{\mathrm{maxCov}}$.

Let a directed path from the start vertex to the end vertex be called a *diploid path*, $D = (D_1 = start, D_2, \ldots, D_{2n+1} = end)$. The graph is naturally organized by the columns of $\mathbf{M}$, so that $D_{2j} = \left(B_j^1, B_j^2\right) \in \mathbf{B_j}$ and $D_{2j+1} = \left(C_{j+1}^1, C_{j+1}^2\right) \in \mathbf{C_j}$ for all $0 < j \leq n$. Let $B_D = \left(B_D^1, B_D^2\right)$ denote a pair of sets, where $B_D^1$ is the union of the first parts of the vertices of $D_2, \ldots, D_{2n+1}$ and, similarly, $B_D^2$ is the union of second parts of the vertices of $D_2, \ldots, D_{2n+1}$.

$B_D^1$ and $B_D^2$ are disjoint because otherwise there must exist a pair of vertices within $D$ that are incompatible, which is easily verified to be impossible. Further, because $D$ visits a vertex for every column of $\mathbf{M}$, it follows that the sum of the cardinalities of these two sets is $m$. $B_D$ is therefore a bipartition of the rows of $\mathbf{M}$ which we call a *diploid path bipartition*.

**Lemma 1** *The set of diploid path bipartitions is the set of bipartitions of the rows of* $\mathbf{M}$ *and each diploid path defines a unique diploid path bipartition.*

*Proof* We first prove that each diploid path defines a unique bipartition of the rows of $\mathbf{M}$. For each column $j$ of $\mathbf{M}$, each vertex $B_j \in \mathbf{B_j}$ is a different bipartition of the same set of active rows. $B_j$ is by definition compatible with a diploid path bipartition of a diploid path that contains it and incompatible with every other member of $\mathbf{B_j}$. It follows that for each column $j$, two diploid paths with the same diploid path bipartition must visit the same node in $\mathbf{B_j}$, and, by identical logic, the same node in $\mathbf{C_j}$, but then two such diploid paths are therefore equal.

There are $2^m$ partitions of the rows of $\mathbf{M}$. It remains to prove that there are $2^m$ diploid paths. By the structure of the graph, the set of diploid paths can be enumerated backwards by traversing right-to-left from the end vertex by depth-first search and exploring each incoming edge for all encountered nodes. As stated previously, the only vertices with indegree greater than one are for all $j$ the members of $\mathbf{C_j}$, and each member of $\mathbf{C_j}$ has the same indegree. For all $j$, the indegree of $C_j$ is clearly $2^{|C_j|-|B_j|}$: two to the power of the number of number of active, terminal rows at column $j$. The number of possible paths must therefore be $\prod_{j=1}^{n} 2^{|C_j|-|B_j|}$. As each row is active and terminal in exactly one column, we obtain $m = \sum_j |C_j| - |B_j|$ and therefore:

$$2^m = \prod_{j=1}^{n} 2^{|C_j|-|B_j|}$$

□

### A hidden Markov model for genotype and diplotype inference

In order to infer diplotypes, we define a hidden Markov model which is based on $G$ but additionally represents all possible genotypes at each genomic site (i.e., in each B column). To this end, we define the set of states $\mathbf{B_j} \times \Sigma_j \times \Sigma_j$, which contains a state for each bipartition of the active rows at position $j$ and all possible assignments of alleles in $\Sigma_j$ to the two partitions. Additionally, the HMM contains a hidden state for each bipartition in $\mathbf{C_j}$, exactly as defined for $G$ above. Transitions between states are defined by the compatibility relationships of the corresponding bipartitions as before. This HMM construction is illustrated in Fig. 8.

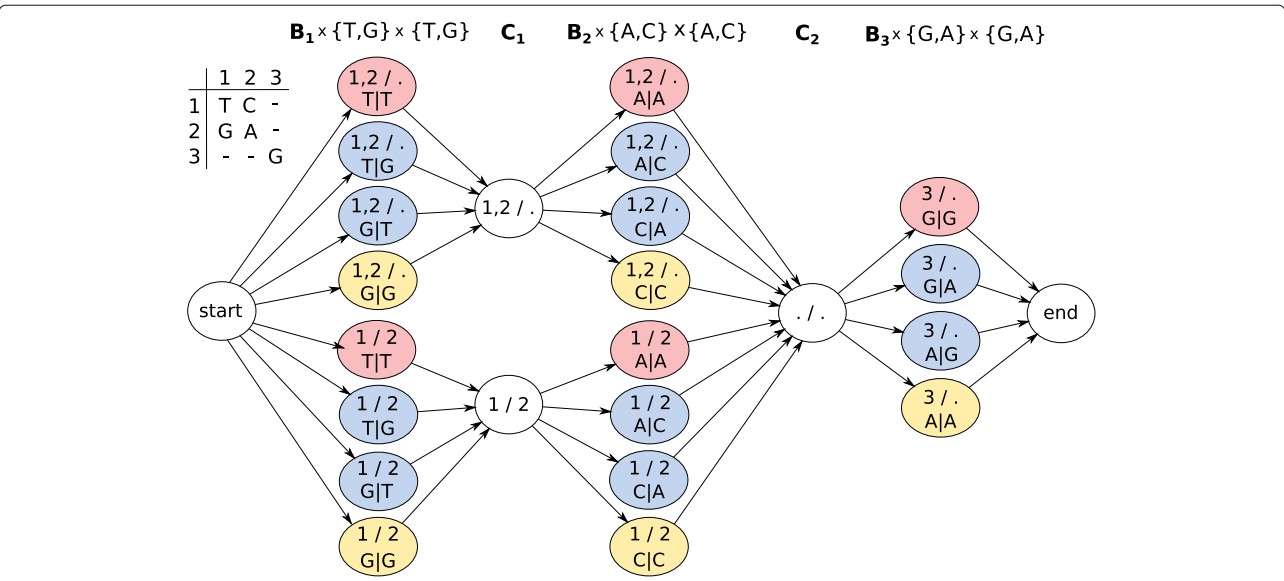

**Fig. 8** Genotyping HMM. Colored states correspond to bipartitions of reads and allele assignments at that position. States in $C_1$ and $C_2$ correspond to bipartitions of reads covering positions 1 and 2 or 2 and 3, respectively. In order to compute genotype likelihoods after running the forward-backward algorithm, states of the same color have to be summed up in each column

For all $j$ and all $C_j \in \mathbf{C_j}$, each outgoing edge has transition probability $P(a_1, a_2) = \sum_{Z_j} P(a_1|Z_j)P(a_2|Z_j)P(Z_j)$, where $(B_j, a_1, a_2) \in \mathbf{B_j} \times \Sigma_j \times \Sigma_j$ is the state being transitioned to. Similarly, each outgoing edge of the start node has transition probability $P(a_1, a_2)$. The outdegree of all remaining nodes is 1, so these edges have transition probability 1.

The start node, the end node, and the members of $\mathbf{C_j}$ for all $j$ are silent states and hence do not emit symbols. For all $j$, members of $\mathbf{B_j} \times \Sigma_j \times \Sigma_j$ output the entries in the $j$th column of $\mathbf{M}$ that are different from "−." We assume every matrix entry to be associated with an error probability, which we can compute from $P(\mathbf{M}_{ij}|H_j^k)$ defined previously. Based on this, the probability of observing a specific output column of $\mathbf{M}$ can be easily calculated.

### Computing genotype likelihoods

The goal is to compute the genotype likelihoods for the possible genotypes for each variant position using the HMM defined above. Performing the forward-backward algorithm returns forward and backward probabilities of all hidden states. Using those, the posterior distribution of a state $(B, a_1, a_2) \in \mathbf{B_j} \times \Sigma_j \times \Sigma_j$, corresponding to bipartition B and assigned alleles $a_1$ and $a_2$, can be computed as:

$$P((B, a_1, a_2)|\mathbf{M}) = \frac{\alpha_j(B, a_1, a_2) \cdot \beta_j(B, a_1, a_2)}{\sum\limits_{B' \in \mathcal{B}(A_j)} \sum\limits_{a_1', a_2' \in \Sigma_j} \alpha_j\left(B', a_1', a_2'\right) \cdot \beta_j\left(B', a_1', a_2'\right)} \quad (1)$$

where $\alpha_j(B, a_1, a_2)$ and $\beta_j(B, a_1, a_2)$ denote forward and backward probabilities of the state $(B, a_1, a_2)$ and $\mathcal{B}(A_j)$,

the set of all bipartitions of $A_j$. The above term represents the probability for a bipartition $B = (B^1, B^2)$ of the reads in $A_j$ and alleles $a_1$ and $a_2$ assigned to these partitions. In order to finally compute the likelihood for a certain genotype, one can marginalize over all bipartitions of a column and all allele assignments corresponding to that genotype.

**Example 1** *In order to compute genotype likelihoods for each column of the alignment matrix, posterior state probabilities corresponding to states of the same color in Fig. 8 need to be summed up. For the first column, adding up the red probabilities gives the genotype likelihood of genotype $T/T$, blue of genotype $G/T$, and yellow of $G/G$.*

### Implementations

We created two independent software implementations of this model, one based upon WhatsHap and one from scratch, which we call MarginPhase. Each uses different optimizations and heuristics that we briefly describe.

### WhatsHap implementation

We extended the implementation of WhatsHap ([22], https://bitbucket.org/whatshap/whatshap) to enable haplotype aware genotyping of bi-allelic variants based on the above model. WhatsHap focuses on re-genotyping variants, i.e., it assumes SNV positions to be given. In order to detect variants, a simple SNV calling pipeline was developed. It is based on samtools mpileup [42] which provides information about the

bases supported by each read covering a genomic position. A set of SNV candidates is generated by selecting genomic positions at which the frequency of a non-reference allele is above a fixed threshold (0.25 for PacBio data, 0.4 for Nanopore data), and the absolute number of reads supporting the non-reference allele is at least 3. These SNV positions are then genotyped using WhatsHap.

### Allele detection
In order to construct the alignment matrix, a crucial step is to determine whether each read supports the reference or the alternative allele at each of $n$ given genomic positions. In WhatsHap, this is done based on re-aligning sections of the reads [16]. Given an existing read alignment from the provided BAM file, its sequence in a window around the variant is extracted. It is aligned to the corresponding region of the reference sequence and, additionally, to the alternative sequence, which is artificially produced by inserting the alternative allele into the reference. The alignment cost is computed by using affine gap costs. Phred scores representing the probabilities for opening and extending a gap and for a mismatch in the alignment can be estimated from the given BAM file. The allele leading to a lower alignment cost is assumed to be supported by the read and is reported in the alignment matrix. If both alleles lead to the same cost, the corresponding matrix entry is "–." The absolute difference of both alignment scores is assigned as a weight to the corresponding entry in the alignment matrix. It can be interpreted as a phred scaled probability for the allele being wrong and is utilized for the computation of output probabilities.

### Read selection
Our algorithm enumerates all bipartitions of reads covering a variant position and thus has a runtime exponential in the maximum coverage of the data. To ensure that this quantity is bounded, the same read selection step implemented previously in the WhatsHap software is run before constructing the HMM and computing genotype likelihoods. Briefly, a heuristic approach described in [43] is applied, which selects phase informative reads iteratively taking into account the number of heterozygous variants covered by the read and its quality.

### Transitions
Defining separate states for each allele assignment in $\mathbf{B_j}$ enables easy incorporation of prior genotype likelihoods by weighting transitions between states in $\mathbf{C_{j-1}}$ and $\mathbf{B_j} \times \Sigma_j \times \Sigma_j$. Since there are two states corresponding to a heterozygous genotype in the bi-allelic case (0|1 and 1|0), the prior probability for the heterozygous genotype is equally spread between these states.

In order to compute such genotype priors, the same likelihood function underlying the approaches described in [44] and [45] was utilized. For each SNV position, the model computes a likelihood for each SNV to be absent, heterozygous, or homozygous based on all reads that cover a particular site. Each read contributes a probability term to the likelihood function, which is computed based on whether it supports the reference or the alternative allele [44]. Furthermore, the approach accounts for statistical uncertainties arising from read mapping and has a runtime linear in the number of variants to be genotyped [45]. Prior genotype likelihoods are computed before read selection. In this way, information of all input reads covering a position can be incorporated.

### MarginPhase implementation
MarginPhase (https://github.com/benedictpaten/margin Phase) is an experimental, open source implementation of the described HMM written in C. It differs from the WhatsHap implementation in the method it uses to explore bipartitions and the method to generate allele support probabilities from the reads.

### Read bipartitions
The described HMM scales exponentially in terms of increasing read coverage. For typical 20–60× sequencing coverage (i.e., average number of active rows per column), it is impractical to store all possible bipartitions of the rows of the matrix. MarginPhase implements a simple, greedy pruning and merging heuristic outlined in recursive pseudocode in Algorithm 1.

The procedure computePrunedHMM takes an alignment matrix and returns a connected subgraph of the

---

**Algorithm 1**

> **procedure** COMPUTEPRUNEDHMM(**M**)
> > **if** maxCov $\geq t$ **then**
> > > Divide **M** in half to create two matrices, $\mathbf{M_1}$ and $\mathbf{M_2}$, such that $\mathbf{M_1}$ is the first $\frac{n}{2}$ rows of **M** and $\mathbf{M_2}$ is the remaining rows of **M**.
> > >
> > > $\mathbf{HMM_1} \leftarrow$ computePrunedHMM($\mathbf{M_1}$)
> > > $\mathbf{HMM_2} \leftarrow$ computePrunedHMM($\mathbf{M_2}$)
> > > $\mathbf{HMM} \leftarrow$ mergeHMMs($\mathbf{HMM_1}, \mathbf{HMM_2}$)
> > **else**
> > > Let **HMM** be the read partitioning HMM for **M**.
> > **return** subgraph of **HMM** including visited states and transitions each with posterior probability of being visited $\geq v$, and which are on a path from the start to end nodes.

HMM for **M** that can be used for inference, choosing to divide the input alignment matrix into two if the number of rows (termed *maxCov*) exceeds a threshold *t*, recursively.

The sub-procedure mergeHMMs takes two pruned HMMs for two disjoint alignment matrices with the same number of columns and joins them together in the natural way such that if at each site $i$ there are $\left|\mathbf{B_i^1}\right|$ states in **HMM₁** and $\left|\mathbf{B_i^2}\right|$ in **HMM₂**, then the resulting HMM will have $\left|\mathbf{B_i^1}\right| \times \left|\mathbf{B_i^2}\right|$ states. This is illustrated in Fig. 9. In the experiments used here $t = 8$ and $v = 0.01$.

### Allele supports

In MarginPhase, the alignment matrix initially has a site for each base in the reference genome. To generate the allele support for each reference base from the reads for each read, we calculate the posterior probability of each allele (reference base) using the implementation of the banded forward-backward pairwise alignment described in [46]. The result is that for each reference base, for each read that overlaps (according to an initial guide alignment extracted from the SAM/BAM file) the reference base, we calculate the probability of each possible nucleotide (i.e., { 'A', 'C', 'G', 'T' }). The gaps are ignored and treated as missing data. This approach allows

summation over all alignments within the band. Given the supports for each reference base, we then prune the set of reference bases considered to those with greater than (by default) three expected non-reference alleles. This expectation is merely the sum of non-reference allele base probabilities over the reads. This reduces the number of considered sites by approximately two orders of magnitude, greatly accelerating the HMM computation.

### Substitution probabilities

We set the read error substitution probabilities, i.e., $P\left(\mathbf{M}_{i,j}|H_j^k\right)$, empirically and iteratively. Starting from a 1% flat substitution probability, we generate a ML read bipartition and pair of haplotypes, we then re-estimate the read error probabilities from the differences between the reads and the haplotypes. We then rerun the model and repeat the process to derive the final probabilities. For the haplotype substitution probabilities, i.e., $P\left(H_j^k|Z_j\right)$, we use substitution probabilities of 0.1% for transversions and 0.4% for transitions, reflecting the facts that transitions are twice as likely empirically but that there are twice as many possible transversions.

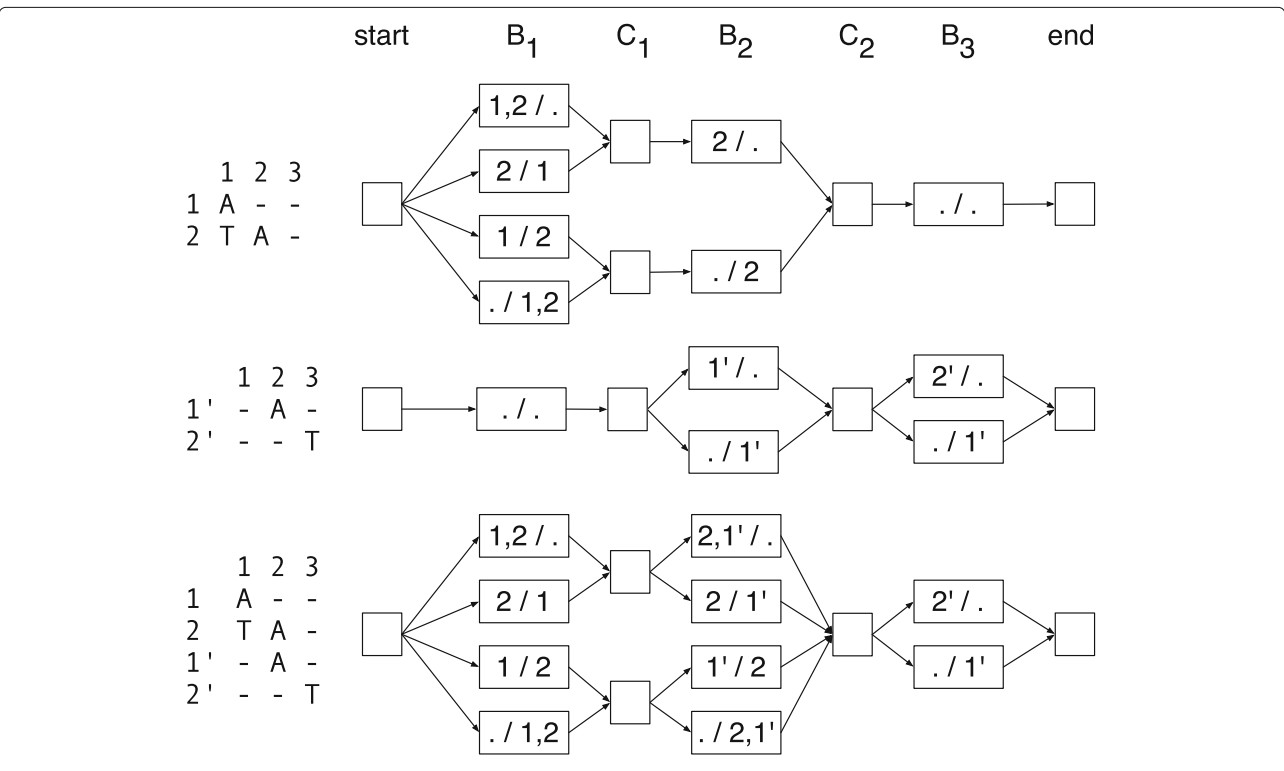

**Fig. 9** The merger of two read partitioning HMMs with the same number of columns. Top and middle: two HMMs to be merged; bottom: the merged HMM. Transition and emission probabilities not shown

## Phase blocks

MarginPhase divides the read partitioning HMMs into phase sets based on the number of reads which span adjacent likely heterozygous sites. The bipartitioning is performed on each of these phase sets individually. MarginPhase's output includes a BAM which encodes the phasing of each read, including which phase set it is in, which haplotype it belongs to, and what of the aligned portion falls into each phase set. Reads which span a phase set boundary have information for both encoded in them.

## Additional files

**Additional file 1:** We provide precision, recall, and genotype concordance of MarginPhase on PacBio and WhatsHap on Nanopore reads. Furthermore, we give the results of cutting and downsampling Nanopore reads. We also provide more detailed phasing statistics and report results on re-typing the GIAB indels. Additionally, we show how precision, recall, and *F*-measure of our callsets behave as a function of read depth and analyze homozygous and heterozygous calls separately. We also provide figures that show the genotyping performance of our methods for different thresholds on the reported genotype qualities. (PDF 695 kb)

**Additional file 2:** Review history. (PDF 220 kb)

## Acknowledgements
We thank the GIAB project for providing the data sets used. In particular, we thank Justin Zook for the helpful discussions on how to use GIAB data and Miten Jain for the help with the nanopore data.

## Funding
This work was supported, in part, by the National Institutes of Health (award numbers: 5U54HG007990, 5T32HG008345-04, 1U01HL137183 R01HG010053, U01HL137183, and U54HG007990 to BP), and by the German Research Foundation (DFG) under award number 391137747 and grants from the W.M. Keck foundation and the Simons Foundation.

## Availability of data and materials
The datasets generated and analyzed during the current study as well as the version of the source code used are available at http://doi.org/10.5281/zenodo.2616973 [47].
MarginPhase and WhatsHap are released as Open Source software under the MIT licence. MarginPhase is available at https://github.com/benedictpaten/marginPhase, and WhatsHap is available at https://bitbucket.org/whatshap/whatshap.

## Authors' contributions
MH, TP, and BP implemented MarginPhase. JE and TM implemented the genotyping module of WhatsHap. All authors contributed to the evaluation and wrote the manuscript. All authors read and approved the final manuscript.

## Review history
The review history is available as Additional file 2.

## Ethics approval and consent to participate
Not applicable.

## Consent for publication
Not applicable.

## Competing interests
The authors declare that they have no competing interests.

## 
## Author details
[1]Center for Bioinformatics, Saarland University, Saarland Informatics Campus E2.1, 66123 Saarbrücken, Germany. [2]Max Planck Institute for Informatics, Saarland Informatics Campus E1.4, Saarbrücken, Germany. [3]Graduate School of Computer Science, Saarland University, Saarland Informatics Campus E1.3, Saarbrücken, Germany. [4]UC Santa Cruz Genomics Institute, University of California Santa Cruz, 95064 Santa Cruz, CA, USA.

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
