## [Review history. (PDF 220 kb) · Genome Biology]

Review History

First round of review

Reviewer 1

Are the methods appropriate to the aims of the study, are they well described, and are necessary controls included?

Yes, the proposed method is well detailed and tested over real data of interest for the contribution.

Are the conclusions adequately supported by the data shown?

Yes.

Are sufficient details provided to allow replication and comparison with related analyses that may have been performed?

I think that the experimental analysis is sufficient even though it could be further improved as suggested in my report.

Does the method perform better than existing methods (as demonstrated by direct comparison with available methods)?

Yes.

Is the method likely to be of broad utility? Is any software component easy to install and use?

The authors have improved and extended a previous tool Whashap with statistical components and additional combinatorial features that allow to solve the genotyping problem and haplotyping from long reads data (Nanopore as well as PacBio reads).

The approach allows to scale to process the human genome compared to a recent reference mentioned in their paper [9]. However, the authors should provide a more detailed discussion in [9].

Is the paper of broad interest to others in the field, or of outstanding interest to a broad audience of biologists?

Yes.

Comments to author:

The paper presents a technique to infer genotypes and haplotypes from long reads using a statistical framework and an algorithmic approach inspired by haplotype assembly. Indeed, the main idea is considering bipartitions of reads associated to the two inferred haplotypes and then estimating the probability of a having a given genotype induced by a given bipartition of long reads.

The authors develop two tools based on the technique. The problem is quite relevant and their technique allows to validate several candidate variants included in reference set of the GIAB project.

The implementation they propose is the result of extending Whatshap, a previous tool for haplotype assembly from long reads with statistical analysis of the bipartitions generated by Whatshap. The work is quite complex from a combinatorial point of view since it combines several approaches: a graph theoretical framework for validating the quality of the alignment of long reads and statistical analysis.

I recommend acceptance of the work.

I only suggest a minor revision that may help to clarify some points of the paper.

Minor revision

- I would like to see a discussion e. g. a statistical analysis on how the novel tools behave in detecting heterozygous versus homozygous sites of the genotypes in the GIAB project.

- You say that the methods rely on computing and estimating bipartitions of the long reads. In other words the approach to infer SNVs validates the values of reads in given positions, i.e. it corrects reads that are first clustered in one of the two partitions, each partition corresponding to a given haplotype that underlies the inferred genotype. From what I know there are other methods in the literature to validate values of reads based on correcting reads and bipartition them to infer haplotypes. For example see [9]. Could you please comment more on the existing literature and approaches for the above purpose, i.e. assembling haplotypes?

- Since you mention work [9] as another approach in the same direction, I would like to see in the Method section a more deep discussion of the existing literature on the topic, as mentioning alternative combinatorial approaches to the genotype assembly or haplotype assembly that lead to solve the phasing problem.

- You say that the approach in [9] does not scale to process the whole genome, but your discussion on how the approach scales does not specify the memory requirements of your approach and how it scaled w.r.t. the dataset characteristics, such as for example coverage or size of the dataset. A more extensive discussion is required.

- On page 9, the first two paragraphs are not very clear to me. You should give more details on what do you mean by saying paving the way for genotyping at intermediate coverage levels. We emphasize that our method operates at coverage levels that preclude the possibility of performing a de novo genome assembly, which, until now, was the most common use of long read data.

- page 9 possibility -> possibility

Reviewer 2

Are the methods appropriate to the aims of the study, are they well described, and are necessary controls included?

Methods are interesting but poorly described.

Are the conclusions adequately supported by the data shown?

Yes.

Are sufficient details provided to allow replication and comparison with related analyses that may have been performed?

Yes.

Does the method perform better than existing methods (as demonstrated by direct comparison with available methods)?

Yes.

Is the method likely to be of broad utility? Is any software component easy to install and use?

As part of the WhatsHap software, yes it will be used. I'm actually a user of the current version of WhatsHap.

Is the paper of broad interest to others in the field, or of outstanding interest to a broad audience of biologists?

Yes: Long read sequencing technologies are becoming popular but there is a lack of statistical methods able to properly process them.

Comments to author:

In this paper, the authors describe and benchmark a new method to perform variant, genotype and haplotype calling from noisy long reads as those generated by sequencing technologies PacBio and Oxford Nanopore. This new method aims at leveraging the ability of long reads to span multiple hetns in order to improve genotype calling and produce reliable haplotype calls. There is indeed a real need for methods able to achieve this task on this data and I appreciate the efforts made by the authors to provide an elegant solution to this problem; i.e. a probabilistic approach able to deal with the uncertainty inherent to this technology. However, I have multiple major concerns about the paper in its current form.

1. Paper organization

Overall, I find the paper really hard to read and understand for two main reasons. First, it took me some time to actually understand that the authors do not compare their new approach to WhatsHap (haplotype assembly methods published in 2015 with a bioRxiv paper from 2016). Instead, they actually compare two different implementations of their HMM method, one stand-alone and the other implemented as a module for WhatsHap. The authors should state this much clearer in their manuscript, for instance, by giving a single name to their method, not two! Overall, this will definitely confuse people. Second, the method description really needs to be rewritten. The description in the main text is very brief (10 lines + 1 line for figure 1B!) and does not give any sense of what the method does, how it does it and what novelty it brings. Please, contrast it better to existing approaches. The formal description at the end of the manuscript is useful for people interested in reproducing their model but clearly not accessible to general readers of Genome Biology. This is why the description in the main text should definitely be improved.

2. Results

You need to explain why MarginPhase gives 96.6% accuracy while whatshap goes up to 99.78% for PacBio genotyping. This is just a massive gap. Why the two implementations you proposed give very different results? Did you study the error mode for each? This is very confusing. At least, you should provide a comparison of the GLs obtained by each approach and clearly interpret/describe the discordances between both.

Along this line, the authors should give guidelines for their two implementations. Which one should I use on my data?

One of the major advantages of having long reads resides in their ability to detect and type SVs. This is one of the key motivations for using these technologies. You should show some results about this. Along the same line, there are some short indels in GIAB if I remember well. Why not showing any results for short indels?

I would like to see the precision/recall for (i) variant discovery, (ii) genotyping and (iii) phasing as a function of coverage. By this I do not mean down-sampling your data as you did, but instead to show how well the method perform as a function of the #reads it uses at each position, i.e. plot figure 3 as a function of coverage.

I would like to see a plot of genotyping accuracy as a function of GL certainty. It would be useful to see if the GLs you compute are well calibrated.

I do not see the point of showing the performance obtained when using two long read approaches. In practice, nobody will do this excepted in few very specific studies. The authors should therefore motivate better why they present this. Something more useful to me would be to assess the performance in terms of phasing/genotyping when you combine deep short Illumina reads with low coverage very long reads.

When you measure phasing errors, how do you deal with the genotyping errors? Do you identify switch errors only at properly typed genotypes?

An optional point would be to show or mention how phasing accuracy is clearly independent from minor allele frequency. This shows to the general readers the advantages of experimental phasing over population based phasing.

Reviewer #1

Comment 1.1: The paper presents a technique to infer genotypes and haplotypes from long reads using a statistical framework and an algorithmic approach inspired by haplotype assembly. Indeed, the main idea is considering bipartitions of reads associated to the two inferred haplotypes and then estimating the probability of a having a given genotype induced by a given bipartition of long reads.

The authors develop two tools based on the technique. The problem is quite relevant and their technique allows to validate several candidate variants included in reference set of the GIAB project.

The implementation they propose is the result of extending Whatshap, a previous tool for haplotype assembly from long reads with statistical analysis of the bipartitions generated by Whatshap. The work is quite complex from a combinatorial point of view since it combines several approaches: a graph theoretical framework for validating the quality of the alignment of long reads and statistical analysis.

I recommend acceptance of the work.

We thank the reviewer for this positive evaluation of our work.

I only suggest a minor revision that may help to clarify some points of the paper.

Minor revision

Comment 1.2: I would like to see a discussion e. g. a statistical analysis on how the novel tools behave in detecting heterozygous versus homozygous sites of the genotypes in the GIAB project.

Response: We thank the reviewer for raising this important point. We included an analysis of this in the supplement. The results confirm our expectation that heterozygous sites are more difficult to detect. For the PacBio data set using WhatsHap, for instance, the recall of homozygous sites is 99.28% while it is only 93.77% for heterozygous sites (see Supplementary Tables S3 and S4 for full details). Likewise the precision of detecting homozygous sites is higher (98.27%) than for detecting heterozygous sites (96.78%). So in summary, the F1 score is 98.77% for homozygous and 95.25% for heterozygous sites. In Nanopore data using MarginPhase, the difference between our ability to characterize homozygous vs. heterozygous sites is even more drastic, with an F1 score of 91.26% for homozygous and 72.02% for heterozygous variants, which is probably due to the increased noise levels in Nanopore data.

Comment 1.3: You say that the methods rely on computing and estimating bipartitions of the long reads. In other words the approach to infer SNVs validates the values of reads in given positions, i.e. it corrects reads that are first clustered in one of the two partitions, each

partition corresponding to a given haplotype that underlies the inferred genotype. From what I know there are other methods in the literature to validate values of reads based on correcting reads and bipartition them to infer haplotypes. For example see [9]. Could you please comment more on the existing literature and approaches for the above purpose, i.e. assembling haplotypes?

Response: We substantially extended our discussion of related work.

First, we point out the connection to the Minimum Error Correction (MEC) problem right at the beginning of the paper and cite five corresponding reviews. While MEC could theoretically serve also for genotyping, all tools that we are aware of make the “all heterozygous” assumption, i.e. they exclusively work on sites previously determined to be heterozygous.

Second, we updated the manuscript with a more detailed review of the method by Guo et al. [9]. In their paper, the authors also evaluate their tool on PacBio data from NA12878 and report substantially worse performance (F1=86.6%) compared to what we observe for our method (F1=97.1%). To reproduce the comparison on exactly the same data set used by us, we attempted to run the application provided by Guo et al., for which only pre-compiled java class files were included in the repository (no source code was present). Unfortunately, we did not succeed in running the tool on our data sets and there was no documentation available. We reached out to the author multiple times, and they indicated they intended to update the repository in response to the problems we reported, but changes have not been made available as of yet (their last response was dated Nov 14). In its present state, the tool does not seem to be usable.

Third, in the meantime, two variant callers based on convolutional neural networks (CNNs) have appeared: a BioRxiv preprint by Luo et al. introduces “Clairvoyante”, a long-read variant caller using CNNs and a paper by Poplin et al. (Nat. Biotech., 2018) introduces “DeepVariant”. The evaluations provided by these authors also compare PacBio variant calls on NA12878 to GIAB benchmark data and are hence comparable to our results. While DeepVariant produces considerably worse results, the results reported for Clairvoyante are comparable to ours. We want to stress, however, that it is presently unclear whether the numbers are overly optimistic estimates since the learning approaches in fact are trained on data from GIAB (for different individual and/or chromosomes). So if the GIAB truth sets should contain any systematic biases or errors, the learning approaches will adopt the same biases and hence will evaluate favorably. While we cannot assess the extent to which this happens, we emphasize that such potential problems are absent from our evaluation. Furthermore, our method additionally reconstruct haplotypes and comes with a statistical model that is readily interpretable.

We now cite the corresponding performance statistics for DeepVariant and Clairvoyante in our paper to put the performance of our method in perspective. Beyond the three tools discussed above, we are not aware of any tools able to generate variant calls from noisy long reads. Hence, we think it is fair to conclude that our approach significantly improves over any published

tools and is on a par with an upcoming tool for which the evaluation has not yet undergone peer review. At the same time, we offer a conceptually novel way of statistically modeling sequencing data from diploid samples, which we envision to serve as the basis for many future tools.

Comment 1.4: Since you mention work [9] as another approach in the same direction, I would like to see in the Method section a more deep discussion of the existing literature on the topic, as mentioning alternative combinatorial approaches to the genotype assembly or haplotype assembly that lead to solve the diplotyping problem.

Response: See above.

Comment 1.5: You say that the approach in [9] does not scale to process the whole genome, but your discussion on how the approach scales does not specify the memory requirements of your approach and how it scaled w.r.t. the dataset characteristics, such as for example coverage or size of the dataset. A more extensive discussion is required.

Response: We apologize for the omission and have updated the manuscript with statistics on memory usage. Guo et al. only ran their method on chr1 of the genome, and did not provide metrics for memory usage or CPU utilization / concurrent threads (only runtime). Unfortunately, we were not able to run this tool to perform our own measurements (see above).

Comment 1.6: On page 9, the first two paragraphs are not very clear to me. You should give more details on what do you mean by saying “paving the way for genotyping at intermediate coverage levels. We emphasize that our method operates at coverage levels that preclude the possibility of performing a de novo genome assembly, which, until now, was the most common use of long read data.”

Response: Thank you. We have provided an analysis in the supplement showing precision and recall as a function of read depth. Furthermore, we have changed the definition of “callable” for our method to include regions which have a minimum depth of 20 instead of 10.

Comment 1.7: page 9 possibilty -> possibility

Response: Corrected.

Reviewer #2

Comment 2.1: In this paper, the authors describe and benchmark a new method to perform variant, genotype and haplotype calling from noisy long reads as those generated by sequencing technologies PacBio and Oxford Nanopore. This new method aims at leveraging

the ability of long reads to span multiple hets in order to improve genotype calling and produce reliable haplotype calls. There is indeed a real need for methods able to achieve this task on this data and I appreciate the efforts made by the authors to provide an elegant solution to this problem; i.e. a probabilistic approach able to deal with the uncertainty inherent to this technology.

Response: We thank the reviewer for her/his general enthusiasm.

Comment 2.2: However, I have multiple major concerns about the paper in its current form.

Response: We are thankful for the points raised. We believe that, by addressing them, we have substantially improved the presentation of methodology and results in our paper. We provide point-by-point responses below.

Comment 2.3: 1. Paper organization

Overall, I find the paper really hard to read and understand for two main reasons. First, it took me some time to actually understand that the authors do not compare their new approach to WhatsHap (haplotype assembly methods published in 2015 with a bioRxiv paper from 2016).

Response: We apologize for creating confusion here. We have updated the introduction to provide more background on haplotype phasing (see response to Comment 1.3) and now also clarify why our existing tools (such as WhatsHap) for haplotyping are not suitable for genotyping. In brief, these tool only work on heterozygous sites given as input.

Comment 2.4: Instead, they actually compare two different implementations of their HMM method, one stand-alone and the other implemented as a module for WhatsHap. The authors should state this much clearer in their manuscript, for instance, by giving a single name to their method, not two! Overall, this will definitely confuse people.

Response: We have updated the text to clarify our decision to provide two implementations. Historically the two implementations emerged from the two groups (Paten/Marschall) working independently on this problem before we joined forces. We considered merging them into one unified tools, but decided against this as the two implementations come with different strength/weaknesses. Each employs different preprocessing steps, which results in each method performing better on one of the sequencing technologies. In addition, we strongly believe that having two independent implementations adds robustness to our results. We now clarify this in the text and provide clear guidance to users by recommending to use MarginPhase for ONT data and WhatsHap for PacBio data. We have restructured the Results section accordingly and now show these combinations in the main text and the less favorable combinations in the supplement.

Comment 2.5: Second, the method description really needs to be rewritten. The description in the main text is very brief (10 lines + 1 line for figure 1B!) and does not give any sense of what

the method does, how it does it and what novelty it brings. Please, contrast it better to existing approaches. The formal description at the end of the manuscript is useful for people interested in reproducing their model but clearly not accessible to general readers of Genome Biology. This is why the description in the main text should definitely be improved.

Response: Thank you for this comment. We addressed it in the following ways: First, we substantially expanded our discussion of related literature, both on related approaches for haplotype phasing as well as on long-read variant calling, in the Background section (see our response to Comment 1.3). Second, we have extended our description of the method in the first section of the Results (Section 2.1, which now is 22 lines long). Third, we have improved the caption of Figure 1b, which was terse indeed. We think that these measures make the novelty of our approach more obvious to readers and provide readers with a high-level understanding of the method, without them having to read through the Method section at the end.

Comment 2.6: 2. Results

You need to explain why MarginPhase gives 96.6% accuracy while whatshap goes up to 99.78% for PacBio genotyping. This is just a massive gap. Why the two implementations you proposed give very different results? Did you study the error mode for each? This is very confusing. At least, you should provide a comparison of the GLs obtained by each approach and clearly interpret/describe the discordances between both.

Response: To elucidate this, we have included in the supplement a detailed analysis of the error profiles of both implementations with respect to homozygous and heterozygous variants (Table S3 and Table S4), the relationship between read depth and accuracy (Figure S3 and Figure S4), and genotype likelihoods (Figure S5). We now emphasize the differences between the two implementations (mostly in preprocessing the data) already in Section 2.1, and point the reader to the corresponding sections in the Methods part where the full details are spelled out.

Comment 2.7: Along this line, the authors should give guidelines for their two implementations. Which one should I use on my data?

Response: We have added a clear recommendation (Section 2.1, Section 4) that WhatsHap should be preferred for PacBio data and MarginPhase should be preferred for ONT data (also see response to Comment 2.4). Furthermore, we have moved the less relevant parts of the analysis to the supplement (i.e. for WhatsHap on ONT data and MarginPhase on PacBio data), so as to provide a clear focus on the configurations we recommend to the user.

Comment 2.8: One of the major advantages of having long reads resides in their ability to detect and type SVs. This is one of the key motivations for using these technologies. You should show some results about this. Along the same line, there are some short indels in GIAB if I remember well. Why not showing any results for short indels?

Response: We agree with the reviewer that indels and SVs are important variant classes that can potentially be accessed using long reads. However, from our previous studies on calling SVs from Oxford Nanopore data (Nat. Comms., 2017, doi:10.1038/s41467-017-01343-4) and from PacBio data (Human Genome Structural Variation Consortium, bioRxiv, 2018, doi: 10.1101/193144), we have to conclude that developing methods for handling SVs properly is a substantial research project in itself and, in our view, outside the scope of the present work. We plan to use the statistical model introduced here as a basis to attack the problem of SV genotyping in the future.

Indels, in particular short tandem repeats (STRs) come with their own challenges and nominating candidate alleles from (indel-error-rich) long reads is particularly difficult (especially using ONT data, where systematic indel errors are quite pronounced). Therefore, due to the error distributions in the input sequencing read types, our present methods are restricted to SNV detection, not including any short insertions or deletions. However, a given variant (consisting of a known REF and ALT allele sequence) can be re-genotyped using WhatsHap, including short indels. To assess this feature, we now additionally provide the results for re-genotyping given indel variants in the supplement.

Comment 2.9: I would like to see the precision/recall for (i) variant discovery, (ii) genotyping and (iii) phasing as a function of coverage. By this I do not mean down-sampling your data as you did, but instead to show how well the method perform as a function of the #reads it uses at each position, i.e. plot figure 3 as a function of coverage.

Response: Thanks you for this suggestion. We provide this analysis in the supplement for both implementations (MarginPhase/WhatsHap) on both read technologies (ONT/PacBio) in Section 5 of the supplement- "Read Depth Analysis" and plot the results in Figures S3 and S4.

Comment 2.10: I would like to see a plot of genotyping accuracy as a function of GL certainty. It would be useful to see if the GLs you compute are well calibrated.

Response: We include these plots in the supplement in Section 7- "Genotype Likelihoods" as Figure S5.

Comment 2.11: I do not see the point of showing the performance obtained when using two long read approaches. In practice, nobody will do this excepted in few very specific studies. The authors should therefore motivate better why they present this. Something more useful to me would be to assess the performance in terms of phasing/genotyping when you combine deep short Illumina reads with low coverage very long reads.

Response: We clarify this better in the text. Specifically we wanted to show that this may be useful for variant confirmation in multi-technology analysis (such as in the NIST GIAB high confidence variant set), and to illustrate that the novel variants confirmed by both technologies are likely to be true variants. We also included more phasing results in Section 2.5 in the

manuscript to emphasize the low switch error rate we observe using both types of long reads.

Comment 2.12: When you measure phasing errors, how do you deal with the genotyping errors? Do you identify switch errors only at properly typed genotypes?

Response: When computing switch errors, we only take variants into account that were genotyped correctly, i.e. all positions genotyped as heterozygous in the truth set and the considered callset. A more in-depth analysis of phasing is provided in the supplement.

Comment 2.13: An optional point would be to show or mention how phasing accuracy is clearly independent from minor allele frequency. This shows to the general readers the advantages of experimental phasing over population based phasing.

Response: Good point. We have added the following sentence to the Discussion: “Furthermore, we note that, unlike approaches using a haplotype reference panel of a population for statistical phasing and/or imputation [28], our approach uses sequencing data of one single individual and its performance is hence independent of allele frequency.”

Reviewer's Responses to Questions

Are the methods appropriate to the aims of the study, are they well described, and are necessary controls included? If not, please specify what is required.

Reviewer #1: yes, the proposed method is well detailed and tested over real data of interest for the contribution

Response: We thank the reviewer for this positive evaluation.

Reviewer #2: Methods are interesting but poorly described.

Response: We hope the updates we have made have clarified this (in particular, see replies to comments 1.3, 2.3, 2.4, 2.5).

Are the conclusions adequately supported by the data shown? If not, please explain.

Reviewer #1: Yes

Reviewer #2: Yes.

Are sufficient details provided to allow replication and comparison with related analyses that may have been performed? If not, please specify what is required.

Reviewer #1: I think that the experimental analysis is sufficient even though it could be further improved as suggested in my report

Response: See our response to Comment 1.3. In short, we attempted to run the analysis done by Guo et al, but the source code is not public, the java class files provided in their github repository had no documentation for analysis of real data, and the author did not fix the problems we faced. We additionally include statistics reported by two recent papers on learning-based variant calling.

Reviewer #2: Yes.

Does the method perform better than existing methods (as demonstrated by direct comparison with available methods)?

Reviewer #1: Yes

Reviewer #2: Yes

Is the method likely to be of broad utility? Is any software component easy to install and use?. Please indicate briefly the novel features and/or advantages of the method, and/or please reference the relevant publications and which methods, if any, it should be compared with.

Reviewer #1: The authors have improved and extended a previous tool WhatsHap with statistical components and additional combinatorial features that allow to solve the genotyping problem and haplotyping from long reads data (Nanopore as well as PacBio reads).

The approach allows to scale to process the human genome compared to a recent reference mentioned in their paper [9].

However, the authors should provide a more detailed discussion on [9].

Response: A more thorough description of the method of Guo et al is included (also see response to Comment 1.3).

Reviewer #2: As part of the WhatsHap software, yes it will be used. I'm actually a user of the current version of WhatsHap.

Response: We are glad to hear that WhatsHap is appreciated by this reviewer. We just hired a dedicated software engineer for the long-term maintenance of WhatsHap.

Is the paper of broad interest to others in the field, or of outstanding interest to a broad audience of biologists? If yes, please explain why.

Reviewer #1: Yes: The paper presents a technique to infer genotypes and haplotypes from long reads using a statistical framework and an algorithmic approach inspired by haplotype assembly. The authors show in the paper the advantage of using long reads in inferring genotypes with high confidence w.r.t. using short reads. Thus their contribution to genotyping as well as in haplotyping is of practical as well as theoretical interest in the field: the results based on the validation of GIAB data are quite good and the tool has the potentiality to be used to build a benchmark for future research.

The main idea of the approach proposed in the paper is considering bipartitions of reads associated to the two inferred haplotypes and then estimating the probability of a having a given genotype induced by a given bipartition of long reads.

The authors develop two tools based on the technique. The problem is quite relevant and their technique allows to validate several candidate variants included in reference set of the GIAB project.

The implementation they propose is the result of extending Whatshap, a previous tool for haplotype assembly from long reads with statistical analysis of the bipartitions generated by Whatshap. The work is quite complex from a combinatorial point of view since it combines several approaches: a graph theoretical framework for validating the quality of the alignment of long reads and statistical analysis, the use of Hidden Markov Models combined with an alignment approach as well as a downsampling technique. In other words the work is not trivial and the experimental analysis requires a deep knowledge of the GIAB data and project.

I recommend acceptance of the work.

I only suggest a minor revision that may help to clarify some points of the paper and provide a more deep discussion of the possible uses of the tool.

Response: We thank the reviewers for this positive assessment and for the many constructive comments.

Reviewer #2: Yes: Long sequencing reads technologies are becoming popular but there is a lack of statistical methods able to properly process them.

Response: We thank the reviewers for confirming the relevance of our approach.

Second round of review

Reviewer 1

The authors improved the manuscript, but still there are some points that need a clarification.

1. Beginning of section 2.1 : I am not sure how novel the statistical framework (based on HMMs) is, since it seems quite similar to, e.g., the Li-Stephens model, for statistical phasing. Please comment on this.
2. Again, these quality metrics, transition/transversion ratio, etc. seem to be a more general version of switch error --- I would be interested in a case where one is high, and the other is very low, for example
3. First paragraph of the Discussion section: the performance is independent of the allele frequency (of reference panels) --- but isn't the performance based purely on the reads that are given ? Could this be subject to bias in the sample, e.g., by low coverage ?
4. Paper [9] is an old study on short reads, and seems fairly unrelated to be worth mentioning, when there are more recent and relevant papers (and tools), such as, e.g., HapCUT, HapCol or ProbHap, for example on haplotype assembly and may be competitors of whatshap.
5. Top of page 20: enumerating all bipartitions of reads covering a variant position, requiring then a preprocessing step [40] is just one way to perform this computation --- while there may be other ways to do so, see the previous comment. I suggest to clarify this point and extend the references in the introductions, since there are other tools for computing bipartitions of long reads in haplotype assembly.
6. While I understand the point that Figure 1 is trying to convey, I think the example could be improved, for example, it is clear that considering the reads calls into question the central SNV call (which would otherwise be called as heterozygous G/T), and that calling it homozygous T/T allows two reads (one from each side) to have an error, but this would also be the case when calling it homozygous G/G, but also heterozygous G/T, in fact. Maybe there is another piece of information that is missing ? For example, if it is more likely to erroneously call a T as a G, than vice versa, this would fully explain this case

Reviewer 2

The authors successfully addressed multiple of my comments. However, for some others, some clarifications are needed:

1. Split 2.4 in two sub-sections to make things clear. One is about detecting variant sites, the other about calling genotypes at these identified positions. Introduce the two concepts clearly.
2. Lines 192-196. This should appear latter in the manuscript when genotyping is discussed.
3. Lines 214-218. Am I right to say that the genotype concordance reported here is the same metric than

the precision reported in Tables S3 and S4 (see comment 6)? If yes, why do we have 99.79% overall and two smaller values when stratified by hom/het (i.e. 98.27% and 96.78%). The same applies to nanopore results. How come 98.02% overall gives 99.23% and 66.47% at hom and het, respectively, unless you have 30 times more hom alt in the data than hets! Please, reconciliation all these numbers!

4. Lines 208-218. The performance at hets should be reported here and not only in the supplementary. This is a crucial piece of information.

5. Lines 220-226. Same applies here, some care is needed. The low switch error has to be put in perspective of the calling performance at hets, i.e.

6. Metric names need to be consistent and described at first use. People will be confused between (i) precision/recall, (ii) genotype concordance, (iii) sensitivity (see additional file, section 6).

7. Section 7 of the supplementary. There are major issues here. First, I do not understand the marginPhase plots. There are basically 3 dots (0, 20, 50) and the remaining at 0 while in the text it is mentioned that the marginPhase's score goes up to 100. Second, why the most certain GLs for WhatsHap on Nanopore are fully discordant with the truth? Third, in my original comment, I referred to calibration, i.e. are the un-phred GLs well calibrated? I mean those coming from the forward-backward HMM pass. In other words, are all genotypes with a posterior of 0.8 correct in 80% of the cases? Phred scaling the GLs here makes the interpretation tricky.

8. Lines 305-311. The 10x difference in running times between two independent implementations and the overall running times need to be discussed.

9. Why Table S1 and S2 appear in the middle of section 5 of the supp? Are they not irrelevant to this section?

10. Opinion. I still think that the packaging of the method can be greatly improved. For future work, I strongly suggest to produce a single software package that can be tuned for specific type of data using simple options (e.g. --pacbio or --nanopore).

Reviewer #1

Comment: The authors improved the manuscript, but still there are some points that need a clarification.

Response: We thank the reviewer for this assessment and respond in detail on how we addressed the residual points below.

Comment: 1. Beginning of section 2.1 : I am not sure how novel the statistical framework (based on HMMs) is, since it seems quite similar to, e.g., the Li-Stephens model, for statistical phasing. Please comment on this.

Response: We now include comments in both section 2.1 and the discussion highlighting the differences between our model and Li-Stephens, specifically that our model uses only read data and does not rely on a model of haplotypes within a population. The Li-Stephens model does not reason about sequencing reads and, in particular, not about partitioning them by haplotypes. In our view, the two models are therefore quite different.

Comment: 2. Again, these quality metrics, transition/transversion ratio, etc. seem to be a more general version of switch error --- I would be interested in a case where one is high, and the other is very low, for example

Response: We have clarified in the text what specifically we mean by the various statistics (ie, switch error, precision/recall, and genotype concordance) in a subsection called "Evaluation Statistics". While the switch error rate pertains to evaluating the haplotype phase between pairs of neighboring variants, the Ti/Tv is a statistic on the nature of the SNVs we detected (and does not reflect the correctness of haplotype phasing). We include the Ti/Tv ratio as a quality control to show that the variants we predict behave as expected with respect to biological processes that determine mutation rates.

Comment: 3. First paragraph of the Discussion section: the performance is independent of the allele frequency (of reference panels) --- but isn't the performance based purely on the reads that are given? Could this be subject to bias in the sample, e.g., by low coverage?

Response: Yes, our method is purely based on sequencing reads. We had inserted the statement that the results are independent of allele frequency based on a suggestion by Reviewer 2 in the first round of revisions. We now edited it once more to make the situation very clear. It now reads: "Furthermore, we note that, unlike approaches using a haplotype reference panel of a population for statistical phasing and/or imputation [31], our approach only uses sequencing data from the individual, hence its performance does not rely on the allele frequency within a population."

While being independent of population allele frequency, indeed the performance of our method does depend on the coverage of the sequencing data, which is investigated in Section 5 of the supplement.

Comment: 4. Paper [9] is an old study on short reads, and seems fairly unrelated to be worth mentioning, when there are more recent and relevant papers (and tools), such as, e.g., HapCUT, HapCol or ProbHap, for example on haplotype assembly and may be competitors of whatshap.

Response: We have added citations to HapCUT and HapCol (ProbHap is already in our citation list), but find that they all attempt to efficiently solve the MEC problem as we have described in our introduction. We maintain the citation of [9] as it focuses on the algorithmic foundations of computational approaches for haplotyping, rather than on short reads.

Comment: 5. Top of page 20: enumerating all bipartitions of reads covering a variant position, requiring then a preprocessing step [40] is just one way to perform this computation --- while there may be other ways to do so, see the previous comment. I suggest to clarify this point and extend the references in the introductions, since there are other tools for computing bipartitions of long reads in haplotype assembly.

Response: Thanks, as mentioned above, we have expanded our citations to include the more recent tools mentioned. However, we did not aim to review all the literature on phasing, especially because we focus on genotyping more in this paper. Note that we cite reviews on haplotyping literature previously written by others [10-11] and ourselves [12].

Comment: 6. While I understand the point that Figure 1 is trying to convey, I think the example could be improved, for example, it is clear that considering the reads calls into question the central SNV call (which would otherwise be called as heterozygous G/T), and that calling it homozygous T/T allows two reads (one from each side) to have an error, but this would also be the case when calling it homozygous G/G, but also heterozygous G/T, in fact. Maybe there is another piece of information that is missing ? For example, if it is more likely to erroneously call a T as a G, than vice versa, this would fully explain this case.

Response: Our intention was to give a simple example which highlights that haplotype information can help identify errors in sequencing (very common in long reads) and can help to expose the corresponding uncertainties when determining genotypes. We agree that the wording was confusing, so we have simplified the text and only claim that the haplotype information shows that there must be read errors at the center locus.

Reviewer #2

Comment: The authors successfully addressed multiple of my comments. However, for some others, some clarifications are needed:

Comment: 1. Split 2.4 in two sub-sections to make things clear. One is about detecting variant sites, the other about calling genotypes at these identified positions. Introduce the two concepts clearly.

Response: We agree that our definitions of the statistics were unclear, and have included in Section 2.2 precise descriptions of how we calculate precision/recall, genotype concordance, and switch error. We hope that this clarifies some of the apparent discrepancies you note below. Additionally, we agree that splitting 2.4 into subsections is an improvement to the formatting and have furthermore included our analysis of phasing accuracy into this structure instead of a separate subsection.

Comment: 2. Lines 192-196. This should appear latter in the manuscript when genotyping is discussed.

Response: We have moved the references to the Supplement so that they are adjacent to the statistics we discuss. We report the homozygous and heterozygous error rates in our primary manuscript (as you suggest below) now. However, since our evaluation of genotype concordance relates to identification of HETs and HOMs (as opposed to allelic accuracies), we feel that it still belongs with our reporting of precision/recall instead of at genotype concordance.

Comment: 3. Lines 214-218. Am I right to say that the genotype concordance reported here is the same metric than the precision reported in Tables S3 and S4 (see comment 6)? If yes, why do we have 99.79% overall and two smaller values when stratified by hom/het (i.e. 98.27% and 96.78%). The same applies to nanopore results. How come 98.02% overall gives 99.23% and 66.47% at hom and het, respectively, unless you have 30 times more hom alt in the data than hets!? Please, reconciliation all these numbers!

Response: The metric we describe as “genotype concordance” is different from the precision/recall metric. We hope that by describing how we calculate genotype concordance (in Section 2.2), this apparent inconsistency is resolved. In short, we use this statistic to describe how well we identify whether a variant is heterozygous or homozygous, and we only include variant calls with correctly identified sites and alleles during its determination.

Comment: 4. Lines 208-218. The performance at hets should be reported here and not only in the supplementary. This is a crucial piece of information.

Response: As mentioned above, we have added these numbers into the main body of the manuscript, and have added brief commentary on the reason for the drop in performance for heterozygous variants.

Comment: 5. Lines 220-226. Same applies here, some care is needed. The low switch error has to be put in perspective of the calling performance at hets, i.e.

Response: Again, we hope that the description of how we calculate switch errors clarifies this, and note that when calculating switch errors we only evaluate at sites we have correctly identified as true HETs. This is so we can separately evaluate genotyping and phasing performance. We use the knowledge that these are true HETs and so genotyping performance does not affect this statistic.

Comment: 6. Metric names need to be consistent and described at first use. People will be confused between (i) precision/recall, (ii) genotype concordance, (iii) sensitivity (see additional file, section 6).

Response: We have updated the text to be consistent.

Comment: 7. Section 7 of the supplementary. There are major issues here. First, I do not understand the marginPhase plots. There are basically 3 dots (0, 20, 50) and the remaining at 0 while in the text it is mentioned that the marginPhase's score goes up to 100. Second, why the most certain GLs for WhatsHap on Nanopore are fully discordant with the truth? Third, in my original comment, I referred to calibration, i.e. are the un-phreded GLs well calibrated? I mean those coming for the forward-backward HMM pass. In other words, are all genotypes with a posterior of 0.8 correct in 80% of the cases? Phred scaling the GLs here make the interpretation Tricky.

Response: We thank the reviewer for this comment. We investigated the drastic drop showing in the plot and found that we accidentally included a point that had no genotyped variants (which was plotted at 0,0); we have removed this point from the plots. The discrepancy between the number of points for WhatsHap and MarginPhase is due to different maximum reported GL. We furthermore included a new plot (Figure S6) to this section of the supplement to show the calibration of our GLs without Phred scaling the values.

Comment: 8. Lines 305-311. The 10x difference in running times between two independent implementations and the overall running times need to be Discussed.

Response: We have described in more detail the causes of the discrepancies between runtimes and memory usage. In short, marginPhase performs a time-intensive realignment of all reads before haplotyping, and furthermore considers roughly 10x the number of reference sites during our evaluation.

Comment: 9. Why Table S1 and S2 appear in the middle of section 5 of the supp? Are they not irrelevant to this section?

Response: Fixed, this was due to improper LaTeX formatting.

Comment: 10. Opinion. I still think that the packaging of the method can be greatly improved. For future work, I strongly suggest to produce a single software package that can be tuned for specific type of data using simple options (e.g. --pacbio or --nanopore).

Response: We thank the reviewer for this comment. We agree and, in fact, we are working towards an implementation that can handle both types of data and plan to include such functionality in future releases of WhatsHap.